# MANTRA: THE MANIFOLD TRIANGULATIONS ASSEMBLAGE

**Rubén Ballester**[*,1,2]  **Ernst Röell**[*,2,3,4]  **Daniel Bīn Schmid**[*,2,3,4]  **Mathieu Alain**[*,5]
**Carles Casacuberta**[1]  **Sergio Escalera**[1,6]  **Bastian Rieck**[2,3,4]

[*]These authors contributed equally to this work
[1]Departament de Matemàtiques i Informàtica, Universitat de Barcelona, Spain
[2]AIDOS Lab, University of Fribourg, Switzerland
[3]Institute of AI for Health, Helmholtz Munich, Germany
[4]Technical University of Munich, Germany
[5]Centre for Artificial Intelligence, University College London, UK
[6]Computer Vision Center, Spain

## ABSTRACT

The rising interest in leveraging higher-order interactions present in complex systems has led to a surge in more expressive models exploiting higher-order structures in the data, especially in topological deep learning (TDL), which designs neural networks on higher-order domains such as simplicial complexes. However, progress in this field is hindered by the scarcity of datasets for benchmarking these architectures. To address this gap, we introduce MANTRA, the first large-scale, diverse, and intrinsically higher-order dataset for benchmarking higher-order models, comprising over 43,000 and 250,000 triangulations of surfaces and three-dimensional manifolds, respectively. With MANTRA, we assess several graph- and simplicial complex-based models on three topological classification tasks. We demonstrate that while simplicial complex-based neural networks generally outperform their graph-based counterparts in capturing simple topological invariants, they also struggle, suggesting a rethink of TDL. Thus, MANTRA serves as a benchmark for assessing and advancing topological methods, paving the way towards more effective higher-order models.

## 1 INTRODUCTION

Success in machine learning is commonly measured by a model's ability to solve tasks on benchmark datasets. While researchers typically devote a large amount of time to build their models, less time is devoted to data and its curation. As a consequence, *graph learning* is facing some issues in terms of reproducibility and wrong assumptions, which serve as obstructions to progress. An example of this was recently observed while analyzing long-range features: additional hyperparameter tuning resolves performance differences between message-passing (MP) graph neural networks on one side and graph transformers on the other (Tönshoff et al., 2023). In a similar vein, earlier work pointed out the relevance of better taxonomies for existing datasets (Liu et al., 2022), as well as the need for strong baselines, highlighting the fact that *structural* information is not exploited equally by all models (Errica et al., 2020). Recently, new analyses even showed that for some benchmark datasets, as well as their associated tasks, graph information may be detrimental for the overall predictive performance (Bechler-Speicher et al., 2024; Coupette et al., 2025), raising concerns about the future of graph learning as a research field (Bechler-Speicher et al., 2025).

These troubling trends concerning data are accompanied by increased interest in leveraging higher-order structures in data, with new models, usually called *topological models*, extending graph-learning concepts to *simplicial complexes*, i.e., generalizations of graphs that incorporate higher-order relations, going beyond the dyadic relations captured by graphs (Alain et al., 2024a; Bodnar et al., 2021c; Maggs et al., 2024; Ramamurthy et al., 2023; Röell & Rieck, 2024; Yang et al., 2024). Some topological models already incorporate state-of-the-art mechanisms for learning such as message-passing (Gilmer et al., 2017) or transformer layers (Ballester et al., 2024), but adapted to higher-order domains, sometimes outperforming their original counterparts in graph datasets. However, as pointed out in a recent position paper (Papamarkou et al., 2024), there is a dire need for "higher-order datasets," i.e., datasets that contain non-trivial higher-order structures. The scarcity of such datasets impedes the development of reliable benchmarks for assessing (i) the utility of higher-order structures present in data, and (ii) the performance of new models that leverage them, thus potentially eroding trust in topological models among the broader deep learning community.

Some of the currently-available higher-order datasets belong to the realm of networks, complex systems, and the life sciences. Benson et al. (2018) present a rich collection of such datasets, comprising nineteen complex networks enhanced with higher-order information. Similar works have also employed higher-order structures in data; for instance, Tadić et al. (2019) use clique complexes on top on graphs coming from brain imaging data. Similarly, Giusti et al. (2016) propose modeling neural data with simplicial complexes by constructing clique, concurrence (and its dual), and independence complexes on the data. However, most of these datasets are either *annotated* or *derived* from simpler data like graphs or time series. In the case of annotated data, it is unclear whether current non-higher-order (graph) neural networks or algorithms can extract the information contained in the higher-order structures using only annotations on vertices and edges. Similarly, for datasets obtained from simpler data, it is also uncertain whether non-higher-order algorithms can recover the higher-order structural information by reconstructing the processes used to generate these relationships explicitly. To the best of our knowledge, the only publicly-available purely higher-order dataset is the "Torus" dataset proposed in Eitan et al. (2025), which consists of a small number of unions of torus triangulations. Due to the nature of the dataset, the only varying topological property among the samples is the number of connected components of each union, making it hard to assess the true capacity of the models to learn and exploit higher-order structures. The lack of higher-order datasets is also remarked upon in a recent benchmarking paper for topological models (Telyatnikov et al., 2024), which restricts itself to existing graph datasets that were subjected to a variety of *topological liftings*, i.e., methods for endowing graph datasets with higher-order structures (Bernárdez et al., 2024; Jonsson, 2007). It remains unclear whether graph neural network architectures can also learn and take advantage of the information provided by the topological liftings, as they are solely based on the graph structure.

**Contributions.** To address these issues, we present MANTRA, the **man**ifold **tr**iangulations **a**ssemblage, which constitutes the first instance of a large, diverse, and intrinsically higher-order dataset, consisting of triangulations of combinatorial 2-manifolds and 3-manifolds. Along with the data, we provide a list of tasks, as well as a preliminary assessment of the performance of existing methods, both graph-based and higher-order-based. We focus on a subset of tasks concerned with the classification of simplicial complexes according to topological labels, where we can interpret the success of a model by its effectiveness in extracting higher-order topological information. However, these tasks are by no means *exhaustive*, and we believe that the generality offered by MANTRA encourages the emergence of even more demanding tasks. Some of these tasks, such as the prediction or approximation of the Betti numbers from topological data, have been previously studied in learning (Paul & Chalup, 2019) and non-learning (Apers et al., 2023) contexts. A noteworthy aspect of MANTRA is the conspicuous *absence* of any intrinsic vertex or edge features such as coordinates or signals. We believe that this absence renders tasks more topological, as models can only rely on topology, instead of non-topological information contained in features. Moreover, as manifold triangulations are directly related to the topological structure of the underlying manifold, we study to which extent higher-order models are *invariant* to transformations of a triangulation that preserve the topological structure of the associated manifold.

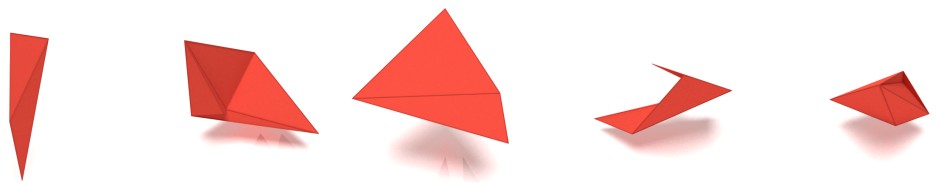

Figure 1: Geometric realizations of some manifold triangulations included in MANTRA. The precise coordinates of vertices in Euclidean space are not geometrically significant; what matters is the topology of the resulting polyhedra. Hence, MANTRA is a *purely combinatorial dataset*.

## 2    DATASET SPECIFICATION

MANTRA contains $43{,}138$ and $250{,}359$ simplicial complexes corresponding to triangulations of closed connected two- and three-dimensional manifolds, respectively, with varying number of vertices, originally curated by Frank H. Lutz and compiled in Lutz (2017). Manifolds have many applications: The configuration space of a robotic arm can be seen as a manifold (e.g., a torus or hyperbolic space, see Jaquier et al., 2022); 3D shapes in geometry processing are triangulated manifolds (Crane & Wardetzky, 2017); physical fields in climate forecasting naturally live on a sphere (Bonev et al., 2023), and the manifold hypothesis argues that high-dimensional data often lies in or close to lower-dimensional manifolds (Fefferman et al., 2016). Throughout the text, we use the term *surface* to refer to a two-dimensional manifold. A *triangulation* of a manifold $M$ is a pair consisting of a simplicial complex K and a homeomorphism between $M$ and the geometric realization of K. For brevity, we use the term triangulation to refer exclusively to the simplicial complex K. See Appendix A.3 for precise definitions and further information.

Table 1: Number of triangulations by manifold dimension ($2\text{-}\mathcal{M}$: 2-manifolds; $3\text{-}\mathcal{M}$: 3-manifolds) and number of vertices $|V|$ in a triangulation.

| $|V|$ | $2\text{-}\mathcal{M}$ | $3\text{-}\mathcal{M}$ |
|---|---|---|
| 4 | 1 | 0 |
| 5 | 1 | 1 |
| 6 | 3 | 2 |
| 7 | 9 | 5 |
| 8 | 43 | 39 |
| 9 | 655 | 1,297 |
| 10 | 42,426 | 249,015 |
| Total | 43,138 | 250,359 |

Triangulations of surfaces and 3-manifolds encode higher-order topological information that cannot be inferred solely from their underlying graphs. Indeed, there exist non-homeomorphic surfaces with identical graph structures. Specifically, for $n > 7$, the complete graph with $n$ vertices triangulates both a connected sum of tori and a connected sum of projective planes, which are non-homeomorphic (Lawrencenko & Negami, 1999). Figure 1 contains examples of geometric realizations of MANTRA triangulations. Table 1 contains the distribution of triangulations in terms of their number of vertices. Each triangulation contains a set of labels based on its dimension. Common labels are the number of vertices of the triangulation, the first three Betti numbers $\beta_0$, $\beta_1$, $\beta_2$, and torsion in homology with integer coefficients. Appendix A.2 provides definitions for these concepts. For triangulations of a Klein bottle $K$, a real projective plane $\mathbb{R}P^2$, a 2-dimensional sphere $S^2$, or a torus $T^2$, the homeomorphism type is included explicitly as a surface label. We additionally specify the top Betti number $\beta_3$ and the homeomorphism type, which can be a 3-sphere $S^3$, a product $S^2 \times S^1$ of a 2-sphere and a circle, or a Möbius-like $S^2$-bundle along $S^1$, denoted by $S^2 \widetilde{\times} S^1$. An exploration of the distributions of labels is made in Appendix A.5.

We make the dataset and benchmark code available via two repositories:

 (i) https://github.com/aidos-lab/MANTRA
(ii) https://github.com/aidos-lab/mantra-benchmarks

These repositories contain (i) the raw and processed datasets, and (ii) the code to reproduce all our results. Detailed hyperparameter settings can be found in Appendix D. Step-by-step instructions on how to set up and execute the benchmark experiments are attached in the README file of the repository. Docker images and workflow, together with package dependencies are included to ensure a unique environment across different machine configurations. Finally, random seeds were used to split the datasets in each run.

The dataset is available in two formats, namely a *raw version* and a version for *PyTorch Geometric*. The raw version currently consists of a pair of compressed files 2_manifolds.json.gz and 3_manifolds.json.gz, containing a JSON list with the triangulations of the corresponding dimension. Each object of the JSON list consists of a set of the following fields, depending on the dimension of the associated triangulation:

| FIELD | TYPE | DESCRIPTION |
|---|---|---|
| id | str | This attribute refers to the original ID of the triangulation as used by Lutz (2017) when compiling the triangulations. This facilitates comparisons to the original dataset if necessary. |
| triangulation | list | A doubly-nested list of the facets of the triangulation. |
| n_vertices | int | The number of vertices in the triangulation. |
| name | str | Homeomorphism type of the triangulation. Possible values are `''`, `'Klein bottle'`,`'RP^2'`,`'S^2'`,`'T^2'` for surfaces, where `''` indicates that the explicit homeomorphism type is not available. For 3-dimensional manifolds, possible values are `'S^2 twist S^1'`,`'S^2 x S^1'`,`'S^3'`. |
| betti_numbers | list | A list of Betti numbers of the triangulation, computed using $R = \mathbb{Z}$, i.e., integer coefficients. |
| torsion_coefficients | list | A list of the torsion subgroups of the triangulation. Possible values are `''`, `'Z_2'`, where an empty string `''` indicates that no torsion is present in that dimension. |
| genus | int | For surfaces, contains the genus of the triangulation. |
| orientable | bool | For surfaces, specifies if the triangulation is orientable or not. |

The PyTorch Geometric (Fey & Lenssen, 2019, PyG) version is available as a Python package that can be installed using the command pip install mantra-dataset. Each example of the dataset is implemented as a PyG Data object, containing the same attributes as JSON objects in the raw version. The main difference with the data in the raw version is that numerical values are stored as PyTorch tensors. Both formats, raw and processed, are versioned using the Semantic Versioning 2.0.0 convention (Preston-Werner) and are also available via Zenodo,[1] thus ensuring *reproducibility* and clear tracking of the dataset evolution.

**Dataset limitations.** In the present version of MANTRA, triangulations are restricted to two- and three-dimensional complexes up to 10 vertices. This may pose a limitation concerning the transferability of our findings to datasets with significantly higher number of vertices per sample, such as fine-grained mesh datasets. While extending the dataset beyond 10 vertices is theoretically possible, it poses substantial storage and computational challenges due to the exponential growth in possible triangulations—for example, there are over 11 million surfaces for triangulations of 11 vertices—and the unavailability of complete enumerations of triangulations for more than 13 vertices, which may potentially lead to incomplete datasets and skewed label distributions. Note that this does not preclude the addition of triangulations with other properties, for instance certain minimality properties (like vertex-transitive triangulations). Additionally, focusing solely on two- and three-dimensional manifolds excludes higher-dimensional triangulations and data, which remain active areas of research. Nevertheless, we believe that MANTRA provides a valuable benchmark for testing

---

[1] https://doi.org/10.5281/zenodo.14103581

higher-order models on the most common types of higher-order structured data, that is, graphs, surfaces, and volumes. Finally, we want to highlight the fact that MANTRA does not (yet) encompass the full spectrum of properties present in real-world data—for example, the large simplicial complex sizes found in complex networks or the geometric information contained in some datasets like the existence of vertex coordinates in meshes. Therefore, although we consider MANTRA a valuable dataset for testing the capabilities of higher-order models, it should be studied in conjunction with other conceptually diverse datasets to better study the capabilities of models.

## 3 EXPERIMENTS

**TL;DR:** We assess twelve state-of-the-art simplicial complex- and graph-based architectures, on various topological prediction tasks such as Betti number homeomorphism type classification, and orientability detection. Our experiments confirm that simplicial complex-based neural networks almost always achieve better results than graph-based ones in extracting the topological invariants mentioned above. However, we also find that the performance of the assessed models may be suboptimal, given that they use the moniker "topological models." In particular, we discover that all model performances deteriorate when applying barycentric subdivisions to the original test datasets, suggesting that the tested models are unable to learn topologically invariant functions.

Sections 3.1 and 3.2 presents the comprehensive experimental design for MANTRA, outlining the key scientific questions addressed. Section 3.3 provides a detailed analysis of the experimental results.

### 3.1 MAIN EXPERIMENTS

In this section, we demonstrate MANTRA's effectiveness as a comprehensive benchmark for higher-order models. Leveraging the extensive set of labels and triangulations available, our experiments are designed to address the following critical research questions:

**Q1** To what extent are higher-order models needed to perform inference tasks on higher-order domains like simplicial complexes? Are graph-based models enough to successfully capture the full set of combinatorial properties present in the data?

**Q2** Do current neural networks, both graph- and simplicial complex-based, capture topological properties in data? Are they able to predict basic topological invariants such as Betti numbers of simplicial complexes?

**Q3** How invariant are state-of-the-art models to transformations that preserve topological properties of data?

The difference between **Q1** and **Q2**, **Q3** is subtle. Combinatorial information is related to the structure of the data, input values, in our case, simplicial complexes, while topological information is related to properties that are invariant under *topological transformations* of the data. For example, in prediction tasks involving molecules, we expect combinatorial information than topological features, since the structure of a molecule is crucial in predicting properties of the molecule. both types of information are intertwined: to properly compute topological properties of data, we need to consider its the input as explained in Appendices A.2 and A.3. To answer the above questions, we benchmarked twelve models: five graph-based (Fey & Lenssen, 2019), using only zero- and one-dimensional simplices of complexes, and seven simplicial complex-based models, four from the TopoModelX library (Hajij et al., 2024) and three extra cellular models, using the full set of simplicial complexes in five different tasks per manifold dimension:

**T1** Predicting the Betti numbers $\beta_i$ for triangulated surfaces and 3-dimensional manifolds.

**T2** Predicting the homeomorphism type of triangulated surfaces.

**T3** Predicting orientability of triangulated surfaces.

These tasks are not challenging when viewed in the context of algebraic topology, since explicit algorithms for all of them are known. However, we are interested in analyzing to what extent a neural network can provide (approximate) solutions here, with the goal of ultimately addressing more challenging tasks in (computational) algebraic topology. To address the high proportion of surfaces without explicitly assigned homeomorphism type, we duplicated the experiments on both the full set of surfaces and the subset of surfaces with known type. Throughout the paper, we denote by $2\text{-}\mathcal{M}^0$, $2\text{-}\mathcal{M}^0_H$, and $3\text{-}\mathcal{M}^0$ the full set of surfaces, the set of surfaces with known homeomorphism type, and the full set of 3-manifolds, respectively.

**Models.** The graph-based models benchmarked are the Multi-Layer Perceptron (MLP), the Graph Convolutional Network (Kipf & Welling, 2017, GCN), the Graph Attention Network (Veličković et al., 2018, GAT), the Graph Transformer (Shi et al., 2021, UniMP), and the Topology Adaptive Graph Convolutional Network (Du et al., 2017, TAG). The simplicial complex-based benchmarked models are the Simplicial Attention Network (Giusti et al., 2022, SAN), three convolution-based simplicial neural networks previously benchmarked in Telyatnikov et al. (2024) and introduced in Yang et al. (2022, SCCN), Yang & Isufi (2023, SCCNN), and Wu et al. (2024, SCN), respectively, the cellular message passing from (Bodnar et al., 2021b, CellMP), the cellular transformer (Ballester et al., 2024, CT), and the Differentiable Euler Characteristic Transform (Röell & Rieck, 2024, DECT). Note that except for the MLP model, the graph and cellular transformers, and the DECT, all models implement some variant of a (higher-order) message-passing paradigm (Hajij et al., 2023; Papillon et al., 2024). More information about the models can be found in Appendix C.

**Features.** All twelve models assume that simplicial complexes are equipped with feature vectors on top of a subset of the simplices. The feature vectors for graph-based models and DECT are either: (1) scalars randomly generated, (2) degrees of each vertex, (3) degree one-hot encodings of each vertex. For the rest of simplicial complex-based models, the feature vectors are either: (1) eight-dimensional vectors generated randomly, (2) number of upper-adjacent neighbors (upper-connectivity index) of each simplex of dimensions lower than the dimension of the simplicial complex and number of lower-adjacent neighbors (lower-connectivity index) for simplices of the same dimension as the simplicial complex. By definition, two simplices are upper-adjacent, and both are upper-adjacent neighbors of the other, if they share a coface of one dimension higher. Similarly, two simplices are lower-adjacent if they share a face of one dimension lower.

**Training details.** In total, our experiments span 240 training results across various tasks, feature generation, and models. To ensure fairness, all configurations use the same learning rate of 0.01 and the same number of epochs of 6; we observe that graph-based models already overfit after a single epoch, though. Hyperparameters for graph-based models were mostly extracted from the default examples from PyTorch Geometric, while hyperparameters for simplicial complex-based models were set to values similar to the ones from the TopoBenchmarkX paper (Telyatnikov et al., 2024), specially for the already benchmarked SCCN, SCCNN, and SCN models. Hyperparameter details can be found in Appendix D. To mitigate the effects of training randomness, we re-ran each experiment five times and considered both the best and the mean (together with standard deviation) performance obtained across these runs for each model and initialisation of features. Due to the high imbalance in the datasets for most labels, we performed stratified train/validation/test splits for each task individually, with 60/20/20 percentage of the data for each split, respectively. Splits were generated using the same random seed for each run, ensuring that the same splits are used across all configurations. All models were trained using the Adam optimizer.

**Loss and metric functions.** Each task (T1, T2, T3) was treated as a classification task during testing. We report the area under the ROC curve (AUROC) (Bradley, 1997) as performance metric, which is standard for imbalanced classification problems, on all tasks except for predicting $\beta_0$, where we report accuracy due to the fact that we only have the label 1, as all our triangulations correspond to connected manifolds. For both the homeomorphism type and orientability tasks, we train the models using the standard cross-entropy loss for classification problems. We also experimented with weighting the cross-entropy loss to penalize

mispredictions in under-represented classes more heavily, but we did not obtain improvements. To avoid increasing the computational complexity of our experiments, we chose not to implement more involved methods for handling the class imbalances and leave this issue for future work. For Betti number prediction, we approached training as a multivariate regression task, since Betti numbers can theoretically be arbitrarily large. Our loss function in this case was the mean squared error, and the Betti number prediction was obtained by rounding the model outputs to the nearest integer.

## 3.2 BARYCENTRIC SUBDIVISION EXPERIMENTS

The previous experiments try to answer **Q1** and **Q2**: if performances are good for simplicial complex-based graph-based ones, then we can conclude that higher-order models are needed to perform inference tasks on domains with higher-order and topological information. By contrast, if performances are good for graph-based models then we can conclude that graph models are enough to capture the full set of combinatorial and topological features present in MANTRA's dataset, questioning the need for higher-order models. However, **Q3** is more subtle. Although it is closely related to **Q2**, **Q3** emphasizes the *invariance* of the models to transformations that preserve the topological properties of the input data, a desirable property for TDL models known as remeshing symmetry (Papamarkou et al., 2024). For example, if a model is well-trained with a dataset containing only triangulations up to a certain number of vertices, we can expect the model to perform correct predictions in new examples that also have at most the maximum number of vertices seen in the training dataset. However, what happens if we try to predict from a *refinement* of a manifold triangulation? For instance, barycentric subdivisions increase the (combinatorial) distances between the original vertices in a triangulation, and this can be harmful for networks relying on the MP algorithm, since distances determine how many layers are needed to propagate information from one vertex to another. In fact, Horn et al. (2022) showed that MP-based graph neural networks with a small number of layers struggled to obtain good performances on synthetic datasets where the number of cycles and connected components played a crucial role.

To answer **Q3**, we performed an additional evaluation of the models trained on surface tasks with known homeomorphism type for the experiments described in Section 3.1. Particularly, for each task, we evaluated the performance of the trained models on a dataset obtained by performing one barycentric subdivision on each triangulation in the original test dataset, and then we compared the performances of the models on both datasets, original and subdivided. Throughout the text, we denote the subdivided test dataset as $2\text{-}\mathcal{M}_H^1$. We did not analyze barycentric subdivisions of 3-dimensional manifolds due to computational constraints. Also, for these experiments, we leave out the DECT model from the analysis, since the DECT is invariant with respect to barycentric subdivision by construction if used appropiately.

## 3.3 ANALYSIS

Our analysis reports *aggregated results* and focuses primarily on the comparison between graph-based models ($\mathcal{G}$) and simplicial complex-based models ($\mathcal{T}$). Comprehensive results are available in Appendix E. Table 2 presents the mean and standard deviation of the maximum performance achieved by each combination of feature vector initialization and model type across the 5 runs of each task for both graph-based ($\mathcal{G}$) and simplicial complex-based ($\mathcal{T}$) model families, including performances on the barycentric subdivisions of the test triangulations for each experiment run in the set of surfaces with known homeomorphism type, as described in Section 3.2. Notably, our experiments suggest that higher-order MP-based and transformer models are *not invariant* relative to topological transformations and therefore cannot be considered topological in the strictest sense of the term: higher-order models predicting better than random in any task suffer from a performance degradation when testing on the subdivided examples, as shown by the full set of results in sections E.1 to E.2 and tables 2 and 11.

Table 2: Predictive performance of graph- and simplicial complex-based models on surface and 3-manifold tasks. Results for the full set of surfaces ($2\text{-}\mathcal{M}^0$), for the set of surfaces with known homeomorphism type ($2\text{-}\mathcal{M}_H^0$), and for the full set of three-manifolds ($3\text{-}\mathcal{M}^0$) are reported. Additionally, performance metrics for the barycentric subdivision of the test set on the models trained on $2\text{-}\mathcal{M}_H^0$, i.e., $2\text{-}\mathcal{M}_H^1$, are included; see Section 3.2 for details. For each family of models, $\mathcal{G}$ (graph-based) and $\mathcal{T}$ (simplicial complex-based), we report the mean and standard deviation of the maximum performance achieved across five runs by each combination of feature vector initialization and model contained in the family. The tasks reported are prediction of $\beta_0, \beta_1, \beta_2, \beta_3$, prediction of the homeomorphism type, and prediction of orientability. For all tasks except for prediction of $\beta_0$, we report the AUROC metric. For $\beta_0$, we report accuracy. Best average result among both families for each task is in bold. Note that the reported averages and standard deviations are not calculated from individual model performances across different random seeds. Instead, for each model, we selected its best performance achieved across all seeds for each experiment. Then, we aggregated these best performances within each category—graph-based and simplicial complex-based models—to compute the averages and standard deviations reported in the table.

| | | Accuracy | | | AUROC | | |
| DATASET | MODEL FAMILY | $\beta_0$ | $\beta_1$ | $\beta_2$ | $\beta_3$ | HOMEO. TYPE | ORIENTABILITY |
|---|---|---|---|---|---|---|---|
| $2\text{-}\mathcal{M}^0$ | $\mathcal{G}$ | **1.00 ± 0.00** | 0.50 ± 0.00 | 0.50 ± 0.00 | | 0.47 ± 0.01 | 0.50 ± 0.00 |
| | $\mathcal{T}$ | 0.73 ± 0.39 | **0.68 ± 0.16** | **0.59 ± 0.10** | | **0.69 ± 0.18** | **0.56 ± 0.07** |
| $2\text{-}\mathcal{M}_H^0$ | $\mathcal{G}$ | **1.00 ± 0.00** | 0.21 ± 0.00 | 0.50 ± 0.00 | | 0.49 ± 0.01 | 0.50 ± 0.00 |
| | $\mathcal{T}$ | 0.57 ± 0.44 | **0.25 ± 0.03** | **0.52 ± 0.02** | | **0.66 ± 0.13** | **0.52 ± 0.02** |
| $2\text{-}\mathcal{M}_H^1$ | $\mathcal{G}$ | **0.47 ± 0.51** | 0.22 ± 0.00 | 0.50 ± 0.00 | | 0.49 ± 0.04 | 0.50 ± 0.00 |
| | $\mathcal{T}$ | 0.21 ± 0.38 | **0.25 ± 0.02** | **0.51 ± 0.01** | | **0.60 ± 0.10** | **0.51 ± 0.01** |
| $3\text{-}\mathcal{M}^0$ | $\mathcal{G}$ | **1.00 ± 0.00** | 0.23 ± 0.00 | 0.12 ± 0.00 | 0.14 ± 0.00 | | 0.14 ± 0.00 |
| | $\mathcal{T}$ | 0.78 ± 0.41 | **0.25 ± 0.04** | **0.13 ± 0.03** | **0.16 ± 0.03** | | **0.15 ± 0.02** |

Weaknesses in the MP-based models are not a recent phenomenon, as highlighted by oversmoothing (Li et al., 2018) and oversquashing (Alon & Yahav, 2021; Topping et al., 2022), and the MP paradigm has required numerous fixes since its existence (inlcuding, but not limited to, virtual nodes, feature augmentation, and graph lifting). More recently, Eitan et al. (2025) argued that, in many cases, higher-order MP-based models cannot distinguish combinatorial objects based on simple topological properties, and has devised another MP variant to compensate for this.

**Graph-based ($\mathcal{G}$) vs. simplicial complex-based ($\mathcal{T}$) models.** Table 2 together with the full results of Appendix E show that simplicial complex-based models consistently obtain better or equivalent performances predicting non-trivial topological properties of triangulated manifolds, meaning $\beta_1, \beta_2, \beta_3$, orientability, and homeomorphism type. We note that graph models *always* correctly detect the connectivity of triangulations in $2\text{-}\mathcal{M}^0$, $2\text{-}\mathcal{M}_H^0$, and 3-dimensional manifolds, thus predicting $\beta_0$ exactly, while topological models consistently fail to predict connectivity, except for the CT, DECT, and SCCN architectures in our experiments. The fact that some higher-order message passing networks cannot accurately predict connectivity was also found, and theoretically proved, in Eitan et al. (2025, Proposition 4.3). Moreover, although simplicial complex-based models obtain better results overall, these are far from being highly accurate, with averages below 70 for all tasks, a high performance variance across the models in some tasks, and tasks for which simplicial complex-based models obtain a performance similar to a random-guessing strategy. Nonetheless, the best performances obtained by specific simplicial complex-based models, as described in the full results of Appendix E, are promising, achieving excellent AUROC results in some tasks, such as homeomorphism type prediction, where the CT and SCCN models obtained average AUROCs of 91 and 83 and 85 and 80 for the full and known homeomorphism type surface datasets, respectively, and Betti number prediction for the full

set of surfaces, where the CT and SCCN models obtained an average AUROC of 93 when predicting the first Betti number. Overall, the results suggest that higher-order models are indeed necessary to capture topological and higher-order characteristics of data, although several current models are not yet able to do so effectively, partially answering questions **Q1** and **Q2**. Such results were expected, given that one-dimensional structures are insufficient, in principle, to fully characterize the topology of two- or three-dimensional triangulated manifolds, as stated at the beginning of Section 2. However, it is plausible that graph-based networks can accurately classify approximately 50% of homeomorphism types of surfaces, since the underlying graph of a triangulation determines the Euler characteristic, which in turn defines the homeomorphism type up to orientability (see Appendix A.4).

**Orientability.** Predicting orientability turns out to be the most difficult task for graph- and simplicial complex-based models, obtaining performances equivalent to random guessing in most cases for surfaces and performances worse than random for 3-manifolds. Moreover, we do not find significative differences between the performances of predicting the Betti number $\beta_2$ and orientability for surfaces as a binary problem. This is consistent with the similar results obtained for predicting $\beta_2$ and orientability type as a binary classification problem for surface datasets.

**Barycentric subdivisions.** Table 2 shows that the performance of all models decreases when subdividing the triangulations of the test dataset if the models were performing better than random guessing, indicating that the models are not learning the invariance of topological properties with respect to subdivisions transformations that leave topological properties invariant. This is a crucial property that any model dealing with topological domains should have, as real data is often highly variable in terms of combinatorial information and representation, but not in terms of their topology. This phenomenon is particularly evident in mesh datasets, where combinatorial structure varies with resolution. Meshes typically comprise more triangles, yet all or nearly all input data represent triangulations of connected closed surfaces, regardless of resolution. In fact, Papamarkou et al. (2024) posit the capacity of TDL models to capture this invariance, denoted *remeshing symmetry*, as one of the reasons for using topological deep learning models. Our preliminary experimental results challenge this claim, opening the door to a new line of research based on the invariance of input transformations that leave topological properties of the input data unaltered. This is important from the perspective of model nomenclature, as many current topological models can be viewed as graph-based approaches that handle heterogeneous node types, with simplices of different dimensions effectively treated as distinct node categories. We believe that if TDL is to emerge as a distinct research area, it must advance the field by introducing higher-order mechanisms that *cannot* be replicated through straightforward adaptations of existing graph models, effectively moving beyond (unaugmented) message-passing approaches. Also, we believe that some tools like (persistent) homology or MAPPER (Singh et al., 2007), which are at presently not used in our experiments, could potentially be used to address or at least alleviate this issue: The expressivity of persistent homology in the context of graph learning has already been studied, and was proven to provide *complementary information* to traditional message-passing approaches (Ballester & Rieck, 2024; Immonen et al., 2023; Rieck et al., 2019), whereas MAPPER was shown to be an effective graph-pooling strategy (Bodnar et al., 2021a). Beyond these two frameworks, we believe that other techniques, drawing upon geometrical-topological concepts, can address this challenge.

**Experimental limitations.** Although our results challenge the efficiency of state-of-the-art higher-order models to predict topological properties of data and open the door to exciting new research avenues, they must be interpreted with care. For example, we mostly tested message-passing networks in our experiments, leaving aside interesting proposals such as higher-order state-space models (Montagna et al., 2024), combinatorial complex networks (Bodnar et al., 2021b; Hajij et al., 2023), topological Gaussian processes (Alain et al., 2024b) or equivariant higher-order neural networks (Battiloro et al., 2025). Due to computational limitations, training procedures were limited to 6 epochs, model hyperparameters were not necessarily selected optimally, and barycentric subdivisions experiments were limited to one subdivision. A significant computational

bottleneck arose from the implementations of simplicial complex-based models, which processed data noticeably slower than their graph counterparts as observed in Table 7, highlighting the need for more efficient implementations of TDL methods as well as *conceptual* improvements. Despite these limitations, we believe that each of our initially-stated questions should be investigated individually, requiring a broader set of experiments and ablations to be fully answered.

## 4 CONCLUSION

We proposed MANTRA, a higher-order dataset of manifold triangulations that is (i) *diverse*, containing triangulations of surfaces and three-dimensional manifolds with different topological invariants and homeomorphism types, (ii) *large*, with over $43,000$ triangulations of surfaces and $250,000$ triangulations of three-dimensional manifolds, and (iii) *naturally higher-order*, as the triangulations are directly related to the topological structure of the underlying manifold. Using MANTRA, we observed that existing models, both graph-based and higher-order-based, struggle to learn topological properties of triangulations, such as the orientability of two-dimensional manifolds, which was the hardest topological property to predict for surface triangulations, suggesting that new approaches are needed to leverage higher-order structure associated with the topological information in the dataset. However, we also observed that current higher-order models outperform graph-based models in our benchmarks, substantiating the promises of this new trend of higher-order machine-learning models. Regarding invariance, we observed that barycentric subdivision deeply affects the performance of the models, suggesting that current state-of-the-art models *fail to be invariant* to transformations that preserve the topological structure of data, opening an interesting research direction for future work. In the case of MP-based models, this could be potentially related to sensitivity of message-passing to the distances between simplices in simplicial complexes. Another interesting research direction for barycentric subdivisions is their application as inputs to graph neural networks. The induced graph of a barycentric subdivision represents each simplex of the original complex as a vertex, with edges encoding face relationships on the original complex. This structure provides an effective representation of simplicial complexes for graph-based neural architectures, potentially facilitating the processing of higher-order topological information. We hope that MANTRA will serve as a benchmark for the development of new models leveraging higher-order and topological structures in data, and as a reference for the development of new higher-order datasets.

## ACKNOWLEDGMENTS

The authors are indebted to Corinna Coupette, the anonymous reviewers, and the area chair for their helpful comments and for believing in our work. RB, SE, and CC were supported by the Ministry of Science and Innovation of Spain through projects PID2019-105093GB-I00, PID2020-117971GB-C22, and PID2022-136436NB-I00. RB was additionally supported by the Ministry of Universities of Spain through the FPU contract FPU21/00968. RB and CC were also supported by the Departament de Recerca i Universitats de la Generalitat de Catalunya (2021 SGR 00697), and SE was additionally supported by ICREA under the ICREA Acadèmia programme. MA was supported by a Mathematical Sciences Doctoral Training Partnership held by Prof. Helen Wilson, funded by the Engineering and Physical Sciences Research Council (EPSRC), under Project Reference EP/W523835/1. This work has received funding from the Swiss State Secretariat for Education, Research, and Innovation (SERI).

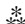

The authors wish to dedicate this work to the memory of *Frank H. Lutz* (1968–2023), who started a collection of triangulations as one of his many research endeavors.

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

# A MATHEMATICAL BACKGROUND

## A.1 SIMPLICIAL COMPLEXES

A *simplicial complex* K is a family of non-empty finite sets such that, if $\sigma \in$ K and $\tau \subseteq \sigma$, then $\tau \in$ K. Each $\sigma \in$ K is called a *simplex* of K, and $\sigma$ is called a *d-dimensional face* or a *d-face* of K if its cardinality is $d+1$. The 0-faces of K are called *vertices* and the 1-faces are called *edges*. We denote by $K^d$ the set of $d$-faces of K, and define the *dimension* of K as the largest $d$ for which $K^d$ is non-empty. A simplicial complex of dimension 1 is called a *graph*.

A *geometric realization* of a simplicial complex K is the union of a collection of affine simplices $\Delta_\sigma$ in a Euclidean space $\mathbb{R}^n$ for some $n \geq 1$, one for each simplex $\sigma \in$ K, where $\sigma$ is mapped bijectively to the vertices of $\Delta_\sigma$, and two affine simplices $\Delta_\sigma$ and $\Delta_\tau$ share a face corresponding to $\sigma \cap \tau$ whenever this intersection is non-empty. Any two geometric realizations of a simplicial complex K are homeomorphic through a face-preserving map.

The *barycentric subdivision* of a simplicial complex K is the simplicial complex Sd(K) obtained by setting its $d$-dimensional faces to be sequences of strict inclusions $\sigma_0 \subset \sigma_1 \subset \cdots \subset \sigma_d$ of simplices of K. It then follows that K and Sd(K) have homeomorphic geometric realizations (Nanda, 2022, Proposition 1.13).

## A.2 SIMPLICIAL HOMOLOGY AND BETTI NUMBERS

Simplicial homology of a simplicial complex K equipped with an order on its set of vertices is defined as follows (Mun, 1984, § 34). Let $R$ be any commutative ring with unit (including the ring of integers $\mathbb{Z}$ or any field). The *chain complex* of K with coefficients in $R$ is a sequence of $R$-modules $(C_n(\mathrm{K}))_{n \in \mathbb{Z}}$ whose elements are formal sums of $n$-simplices of K with coefficients in $R$, i.e.,

$$C_n(\mathrm{K}) = \left\{ \textstyle\sum_{\sigma \in \mathrm{K}^n} a_\sigma \sigma \mid a_\sigma \in R \right\},$$

linked by *boundary homomorphisms* $\partial_n \colon C_n(\mathrm{K}) \to C_{n-1}(\mathrm{K})$ for all $n \in \mathbb{Z}$, given by

$$\partial_n \left( \textstyle\sum_{\sigma \in \mathrm{K}^n} a_\sigma \sigma \right) = \textstyle\sum_{\sigma \in \mathrm{K}^n} a_\sigma \partial_n(\sigma), \qquad \partial_n(\sigma) = \textstyle\sum_{i=0}^{n} (-1)^i (\sigma \smallsetminus \{v_i\}),$$

if $v_0, \ldots, v_n$ are the ordered vertices of $\sigma$. The main property of the boundary homomorphisms is that $\partial_n \circ \partial_{n+1} = 0$ for all $n$, implying that $\mathrm{Im}(\partial_{n+1}) \subseteq \mathrm{Ker}(\partial_n)$ for all $n$. This yields *homology $R$-modules*, defined as quotients $H_n(\mathrm{K}) = \mathrm{Ker}(\partial_n)/\mathrm{Im}(\partial_{n+1})$ for all $n$.

If K is a finite simplicial complex and $R = \mathbb{Z}$, then $H_n(\mathrm{K})$ is a finitely generated abelian group and therefore it decomposes as a direct sum

$$H_n(\mathrm{K}) \cong \mathbb{Z}^{\beta_n} \oplus \mathbb{Z}_{q_1} \oplus \cdots \oplus \mathbb{Z}_{q_t},$$

where $\beta_n$ is the *n-th Betti number* of K, while $q_1, \ldots, q_t$ are prime powers. The sum $\mathbb{Z}_{q_1} \oplus \cdots \oplus \mathbb{Z}_{q_t}$ is the *torsion* subgroup of $H_n(\mathrm{K})$. Examples of Betti numbers are provided in Figure 2. The $n$-th Betti number of a simplicial complex K counts the number of linearly independent $n$-dimensional cavities in a geometric realization of K. In low dimensions, $\beta_0$ is equal to the number of connected components, and $\beta_1$ counts the number of linearly independent loops that are not boundaries of any 2-dimensional region.

## A.3 TRIANGULATED MANIFOLDS

An *n-dimensional manifold* is a second-countable Hausdorff topological space $M$ such that every point of $M$ is contained in some open set, called a *chart*, equipped with a homeomorphism into an open subset of a Euclidean space $\mathbb{R}^n$ (Mun, 1984, § 36). This definition does not include manifolds with boundary, which are not considered in this article. A manifold is called *closed* if its underlying topological space is compact.

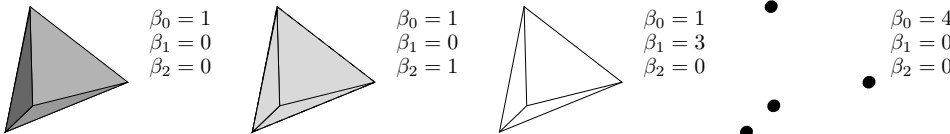

Figure 2: From left to right, four simplicial complexes $K_1$, $K_2$, $K_3$, and $K_4$ with their respective Betti numbers $\beta_0$, $\beta_1$, and $\beta_2$. The $n$-th Betti number indicates the number of $n$-dimensional holes in a geometric realization of a simplicial complex. Here $K_1$ is a solid tetrahedron with $\beta_0 = 1$, $\beta_1 = 0$, and $\beta_2 = 0$, since $K_1$ has only one connected component, no unfilled cycles, and no empty cavity enclosed by 2-faces; $K_2$ is a hollow tetrahedron with $\beta_0 = 1$, $\beta_1 = 0$, and $\beta_2 = 1$ (the difference with $K_1$ is that the triangles of $K_2$ enclose a cavity); $K_3$ is the underlying graph, with $\beta_0 = 1$, $\beta_1 = 3$, and $\beta_2 = 0$, since there is no cavity and there are three linearly independent cycles; $K_4$ consists of four vertices and has $\beta_0 = 4$, $\beta_1 = 0$, and $\beta_2 = 0$, since there are four connected components and no cycles nor cavities.

A collection of charts covering a manifold $M$ is an *atlas* of $M$. A manifold $M$ is called *orientable* if it admits an atlas with compatible orientations in its charts. For a closed $n$-dimensional manifold $M$, orientability is determined by its $n$-th Betti number $\beta_n$, which is nonzero if and only if $M$ is orientable.

A *triangulation* of a manifold $M$ is a simplicial complex whose geometric realization is homeomorphic to $M$. Radó (1925) proved that every surface admits a triangulation (which can be chosen to be finite if the surface is compact), and that any two such triangulations admit a common refinement. Moise (1952) proved that the same facts are true for 3-dimensional manifolds. For dimensions greater than 3, however, there are examples of manifolds that cannot be triangulated.

## A.4 CLASSIFICATION

Closed connected surfaces can be classified, up to homeomorphism, as given by the following list: (i) the two-dimensional sphere $S^2$; (ii) a connected sum of tori $T^2$; (iii) a connected sum of projective planes $\mathbb{R}P^2$. The *genus* of a surface $M$ is defined as zero if $M \cong S^2$ and equal to $g$ if $M$ is a connected sum of $g$ tori or $g$ projective planes. Thus the homeomorphism type of $M$ is determined by its orientability and genus.

The *Euler characteristic* of a finite triangulation of a manifold $M$ is the alternating sum of the numbers of simplices of each dimension. It does not depend on the choice of a triangulation, and it is equal to the alternating sum of the Betti numbers of $M$ (Hatcher, 2002). The Euler characteristic of a closed connected surface $M$ of genus $g$ is equal to $2 - 2g$ if $M$ is orientable and $2 - g$ if $M$ is not orientable.

The underlying graph of a finite triangulation of a closed surface $M$ determines the Euler characteristic $v - e + t$. This is due to the fact that, in any triangulation of $M$, each edge bounds precisely two triangles, so $3t = 2e$. Therefore, the underlying graph of a triangulation of a closed surface $M$ determines the homeomorphism type of $M$ up to orientability. As shown in Lawrencenko & Negami (1999), the torus and the Klein bottle admit triangulations with the same underlying graph.

For manifolds of dimension greater than 2, classification up to homeomorphism is so far unfeasible. In dimension 3, the geometrization theorem (Morgan & Tian, 2007) describes all possible geometries of prime components of closed 3-manifolds. The Euler characteristic does not carry any information about the homeomorphism type in dimension 3, since if $M$ is any odd-dimensional closed manifold then $\chi(M) = 0$ by Poincaré duality (Hatcher, 2002, 3.37). However, the underlying graph of a finite triangulation of a closed 3-manifold determines the number $t$ of triangles and the number $f$ of 3-faces, since $4f = 2t$ and $v - e + t - f = 0$.

## A.5 DISTRIBUTION OF LABELS

Tables 3, 4, 5, and 6 contain statistical information about the distribution of labels in the dataset.

Table 3: Distribution of Betti numbers $\beta_i$ for triangulations of manifolds. Percentages are rounded to the nearest integer and are computed for each pair of manifold dimension (2 or 3) and Betti number. Columns represent values of Betti numbers and contain the number of manifolds with each value.

|  | $\mathcal{M}$ | 0 | 1 | 2 | 3 | 4 | 5 | 6 |
|---|---|---|---|---|---|---|---|---|
| $\beta_0$ | 2-$\mathcal{M}$ | - | 43,138 (100%) | - | - | - | - | - |
|  | 3-$\mathcal{M}$ | - | 250,359 (100%) | - | - | - | - | - |
| $\beta_1$ | 2-$\mathcal{M}$ | 1,670 (4%) | 4,655 (11%) | 14,146 (33%) | 13,694 (32%) | 7,917 (18%) | 1,022 (2%) | 34 (0%) |
|  | 3-$\mathcal{M}$ | 249,225 (100%) | 1,134 (0%) | 0 (0%) | 0 (0%) | 0 (0%) | 0 (0%) | 0 (0%) |
| $\beta_2$ | 2-$\mathcal{M}$ | 39,718 (92%) | 3,420 (8%) | - | - | - | - | - |
|  | 3-$\mathcal{M}$ | 249,841 (100%) | 518 (0%) | - | - | - | - | - |
| $\beta_3$ | 2-$\mathcal{M}$ | - | - | - | - | - | - | - |
|  | 3-$\mathcal{M}$ | 616 (0%) | 249,743 (100%) | - | - | - | - | - |

Table 4: Distribution of torsion subgroups for triangulations of manifolds. Percentages are rounded to the nearest integer and are computed for each pair of manifold dimension and homological degree.

|  | $H_0$ | $H_1$ | | $H_2$ | | $H_3$ |
|---|---|---|---|---|---|---|
| $\mathcal{M}$ | 0 | $\mathbb{Z}_2$ | 0 | $\mathbb{Z}_2$ | 0 | 0 |
| 2-$\mathcal{M}$ | 43,138 (100%) | 39,718 (92%) | 3,420 (8%) | 0 (0%) | 43,138 (100%) | - |
| 3-$\mathcal{M}$ | 250,359 (100%) | 0 (0%) | 250,359 (100%) | 616 (0%) | 249,743 (100%) | 250,359 (100%) |

Table 5: Distribution of genus for triangulations of surfaces. Percentages are rounded to the nearest integer.

| $\mathcal{M}$ | 0 | 1 | 2 | 3 | 4 | 5 | 6 | 7 |
|---|---|---|---|---|---|---|---|---|
| 2-$\mathcal{M}$ | 306 (1%) | 3,593 (8%) | 5,520 (13%) | 11,937 (28%) | 13,694 (32%) | 7,052 (16%) | 1,022 (2%) | 14 (0%) |

Table 6: Distribution of homeomorphism types for triangulations of manifolds. Percentages are rounded to the nearest integer and are computed for each manifold dimension. Surfaces classified as "Other" do not have any explicit homeomorphism type assigned.

| $\mathcal{M}$ | $S^2$ | $\mathbb{R}P^2$ | $T^2$ | $K$ | $S^3$ | $S^2 \times S^1$ | $S^2 \tilde{\times} S^1$ | Other |
|---|---|---|---|---|---|---|---|---|
| 2 - $\mathcal{M}$ | 306 | 1,364 | 2,229 | 4,655 | - | - | - | 34,584 |
| | (1%) | (3%) | (5%) | (11%) | | | | (80%) |
| 3 - $\mathcal{M}$ | - | - | - | - | 249,225 | 518 | 616 | 0 |
| | | | | | (100%) | (0%) | (0%) | (0%) |

# B  DATASET DETAILS

This section provides additional details about the dataset and the design choices involved in its creation.

## B.1  DATA FORMAT

> This section is mostly *information-oriented* and provides a brief overview of the data format, followed by a short example.

As a complement to Section 2 in the main text, we provide an extended description of dataset attributes. Each dataset consists of a list of triangulations, with each triangulation having the following attributes:

- `id` (required, `str`): This attribute refers to the original ID of the triangulation following Lutz (2017). This facilitates comparisons to the original dataset if necessary and simplifies future contributions by other authors.

- `triangulation` (required, `list` of `list` of `int`): A doubly-nested list of the top-level simplices of the triangulation.

- `n_vertices` (required, `int`): The number of vertices in the triangulation. This is **not** the number of simplices.

- `name` (required, `str`): A canonical name of the triangulation, such as `S^2` for the two-dimensional sphere. If no canonical name exists, we store an empty string.

- `betti_numbers` (required, `list` of `int`): A list of the Betti numbers of the triangulation, computed using $\mathbb{Z}$ coefficients. This implies that torsion coefficients are stored in another attribute.

- `torsion_coefficients` (required, `list` of `str`): A list of the torsion coefficients of the triangulation. An empty string `''` indicates that no torsion coefficients are available in that dimension. Otherwise, the original spelling of torsion coefficients is retained, so a valid entry might be `'Z_2'`.

- `genus` (optional, `int`): For 2-manifolds, contains the genus of the triangulation.

- `orientable` (optional, `bool`): Specifies whether the triangulation is orientable or not.

We picked JSON as our underlying data format, since it facilitates exchanging information, extending the dataset, and can be easily processed in all major programming languages. By only ever storing the top-level simplices of a simplicial complex, the dataset can be easily compressed. Moreover, the textual format facilitates tracking changes over different versions of the dataset. The subsequent listing depicts a simple example of the data format; see below for additional design choices.

```
1   [
2     {
3       "id": "manifold_2_4_1",
4       "triangulation": [
5         [1,2,3],
6         [1,2,4],
7         [1,3,4],
8         [2,3,4]
9       ],
10      "dimension": 2,
11      "n_vertices": 4,
12      "betti_numbers": [
13        1,
14        0,
15        1
16      ],
17      "torsion_coefficients": [
18        "",
19        "",
20        ""
21      ],
22      "name": "S^2",
23      "genus": 0,
24      "orientable": true
25    },
26    {
27      "id": "manifold_2_5_1",
28      "triangulation": [
29        [1,2,3],
30        [1,2,4],
31        [1,3,5],
32        [1,4,5],
33        [2,3,4],
34        [3,4,5]
35      ],
36      "dimension": 2,
37      "n_vertices": 5,
38      "betti_numbers": [
39        1,
40        0,
41        1
42      ],
43      "torsion_coefficients": [
44        "",
45        "",
46        ""
47      ],
48      "name": "S^2",
49      "genus": 0,
50      "orientable": true
51    }
52  ]
```

### B.2   DESIGN CHOICES

> This section is *understanding-oriented* and provides additional justifications for our data format.

The datasets are converted from their original (mixed) lexicographical format (Lutz, 2008). A triangulation in lexicographical format could look like this:

```
1  manifold_lex_d2_n6_#1=[[1,2,3],[1,2,4],[1,3,4],[2,3,5],[2,4,5],[3,4,6],
2    [3,5,6],[4,5,6]]
```

A triangulation in *mixed* lexicographical format could look like this:

```
1  manifold_2_6_1=[[1,2,3],[1,2,4],[1,3,5],[1,4,6],
2    [1,5,6],[2,3,4],[3,4,5],[4,5,6]]
```

This format is **hard to parse** and error-prone. Moreover, any *additional* information about the triangulations, including information about homology groups or orientability, for instance, requires additional files. We thus decided to use a format that permits us to keep everything in one place, including any additional attributes for a specific triangulation. A desirable data format needs to satisfy the following properties:

1. It should be easy to parse and modify, ideally in a number of programming languages.
2. It should be human-readable and `diff`-able in order to permit simplified comparisons.
3. It should scale reasonably well to larger triangulations.

After some considerations, we decided to opt for `gzip`-compressed JSON files. JSON is well-specified and supported in virtually all major programming languages out of the box. While the compressed file is *not* human-readable on its own, the uncompressed version can easily be used for additional data analysis tasks. This also greatly simplifies maintenance operations on the dataset. While it can be argued that there are formats that scale even better, they are not well-applicable to our use case since each triangulation typically consists of different numbers of top-level simplices. This rules out column-based formats like Parquet.

> We are open to revisiting this decision in the future. Our current API can be adjusted to accommodate other data formats. End users are *not* interacting with the raw data.

As for the *storage* of the data as such, we decided to keep only the top-level simplices (as is done in the original format) since this substantially saves disk space. The drawback is that the client has to supply the remainder of the triangulation. Given that the triangulations in our dataset are not too large, we deem this to be an acceptable compromise. Moreover, data structures such as simplex trees can be used to further improve scalability if necessary. Finally, our data format includes, whenever possible and available, additional information about a triangulation, including the Betti numbers and a *name*, i.e., a canonical description, of the topological space described by the triangulation. We opted to minimize any inconvenience that would arise from having to perform additional parsing operations.

Overall, this data format remains extensible—permitting additional information about a triangulation or attributes like coordinates—while still benefiting from easy accessibility. We make our code and data publicly available and use Zenodo for long-term archival with DOIs. The most recent version of our dataset is accessible via:

https://doi.org/10.5281/zenodo.14103582

Old versions are archived and can be accessed using our data loader. We hope that this system, while not perfect, may serve as a suitable starting point for other benchmark datasets.

## C  MODEL DETAILS

We provide a brief description of the models used in the experiments.

**Message passing neural networks.**   Most of the models used in the literature for graphs and higher-order structures such as simplicial or cell complexes are based on the message-passing paradigm. For graph and simplicial complexes, these models *pass* messages between *neighboring* nodes or simplices in the graph or complex, updating their features based on the features of their neighbors. Let K be a simplicial complex or a graph seen as a simplicial complex with simplicial features given by a family of maps $\{F_i\}_{i=0}^{\dim K}$ where $F_i \colon K_i \to \mathbb{R}^{d_i}$. A message-passing layer updates the features of a simplex $\sigma$ using the following steps (Papillon et al., 2024):

1. *Selection of neighborhoods*: Given a simplex $\sigma$, we first start by defining sets of neighboring simplices $\{\mathcal{N}_i(\sigma)\}_i$ where the neighborhoods are defined depending on the context. For example, adjacent or incident simplices are two types of neighborhoods that can be defined in an arbitrary simplicial complex. Usually, neighborhoods are defined in the same way for the same dimension of simplices, and each set of neighboring simplices contain simplices of the same dimension.

2. *Message computation*: For each set of neighboring simplices $\mathcal{N}(x)_i$, we compute messages $\{m_{\tau \to \sigma}\}_i$ from the features of the simplices in $\mathcal{N}_i(x)$ and the features of $\sigma$, this is

$$m_{\tau \to \sigma} = M_{\mathcal{N}(x)}(F_{\dim \tau}(\tau), F_{\dim \sigma}(\sigma), \Theta),$$

   where $\Theta$ are the learnable parameters of the layer.

3. *Intra-aggregation*: The messages are aggregated to obtain a single message for each neighborhood $\mathcal{N}_i(x)$, this is

$$m_{\mathcal{N}_i(x)} = \mathrm{Agg}_{\mathcal{N}_i(x)}(\{m_{\tau \to \sigma}\}_{\tau \in \mathcal{N}_i(x)}),$$

   where $\mathrm{Agg}_{\mathcal{N}_i(x)}$ is an permutation invariant aggregation function, for example, a sum, mean, or any other function that aggregates the messages.

4. *Inter-aggregation*: The aggregated messages for the neighborhoods are then aggregated together to obtain a single message for the simplex $\sigma$, this is

$$m_\sigma = \mathrm{Agg}_\sigma(\{m_{\mathcal{N}_i(x)}\}_i),$$

   where $\mathrm{Agg}_\sigma$ is a permutation invariant aggregation function again.

5. *Update*: The message $m_\sigma$ is used to update the features of the simplex $\sigma$, this is

$$F_{\dim \sigma}(\sigma) = \mathrm{Update}(F_{\dim \sigma}(\sigma), m_\sigma, \Theta).$$

For graphs, GCN (Kipf & Welling, 2017), GAT (Veličković et al., 2018), and UniMP (Shi et al., 2021) are examples of message-passing networks. In the case of GCN and GAT, adjacency with self-loops is used as neighborhood sets for nodes, whereas UniMP uses concatenated adjacencies up to a order $k$, meaning that we consider as neighbors of a vertex all the other vertices of the graph at a distance of at most $k$ from the vertex. In the case of GAT, the fundamental difference lie in the message computation, where the message from a simplex $\tau$ to a simplex $\sigma$ depends on a concept of attention, which is computed using the features of $\tau$ and $\sigma$ and a learnable parameter $\Theta$.

In the case of simplicial complexes, SAN (Giusti et al., 2022) and SCN (Wu et al., 2024) use (upper and lower) higher-order Laplacians to define neighborhoods, SCCN (Yang et al., 2022) uses (co)adjacency and incidence structures, and SCCNN (Yang & Isufi, 2023) uses all together.

**Non-message passing neural networks**    Although the message-passing paradigm is predominant in the literature, there are other state-of-the-art models that do not follow this paradigm, such as transformers (Ballester et al., 2024), state-space topological models (Montagna et al., 2024), or TDA-based networks (Horn et al., 2022). In our case, we only select graph and cellular transformers and multi-layer perceptrons (MLP) for comparison. Graph and cellular transformers are based on the original transformer's decoder architecture (Vaswani et al., 2017). Original transformer architectures are permutation-invariant networks that use positional encoding to break the symmetry of the input data by means of localizing the position of each element in the input sequence. In the case of graph and cellular transformers, which do not always have a linear structure as text, positional encodings encode the *position* of the different simplices in the simplicial complex using the combinatorial structure of the complex. Famous positional encodings for graphs are built using eigenvectors of the graph Laplacian and random walks (Müller et al., 2024). For simplicial transformers, preliminary positional encodings are based also on eigenvectors of combinatorial Laplacians, random walks, and graph positional encodings for barycentric subdivisions of the simplicial complexes.

## D  HYPERPARAMETER DETAILS

More information on the meaning of specific hyperparameters can be found in the PyTorch geometric, TopoModelX, CellMP, CT and DECT original implementations.

| GRAPH MODELS & DECT | |
|---|---|
| **GAT** | |
| Hidden neurons | 64 |
| Hidden layers | 4 |
| Readout | Mean |
| Dropout last linear layer | 0.5 |
| Activation last layer | Identity |
| **GCN** | |
| Hidden neurons | 64 |
| Hidden layers | 4 |
| Readout | Mean |
| Dropout last linear layer | 0.5 |
| Activation last layer | Identity |
| **MLP** | |
| Hidden neurons | 64 |
| Hidden layers | 4 |
| Readout | Mean |
| Dropout last linear layer | 0.0 |
| Activation last layer | Identity |
| **TAG** | |
| Hidden channels | 64 |
| Hidden layers | 4 |
| Readout | Mean |
| Dropout last linear layer | 0.5 |
| Activation last layer | Identity |
| **UNiMP** | |
| Hidden channels | 64 |
| Hidden layers | 4 |
| Readout | Mean |
| Dropout last linear layer | 0.5 |
| Activation last layer | Identity |
| **DECT** | |
| Hidden channels | 64 |
| Hidden layers | 3 |
| Number of $\theta$ | 32 |
| Bump steps | 32 |
| $r$ | 1.1 |
| Normalized | True |

| TOPOLOGICAL MODELS | |
|---|---|
| **SAN** | |
| Hidden channels | 64 |
| Hidden layers | 1 |
| $n$-filters | 2 |
| Order harmonic | 5 |
| Epsilon harmonic | 1e-1 |
| Readout | Sum of sums per dimension |
| **SCCN** | |
| Hidden channels | 64 |
| Hidden layers | 2 |
| Maximum rank | 2 |
| Aggregation activation function | Sigmoid |
| Readout | Sum of sums per dimension |
| **SCCNN** | |
| Hidden channels | 64 |
| Hidden layers | 2 |
| Order of convolutions | 1 |
| Order of simplicial complexes | 1 |
| Readout | Sum of sums per dimension |
| **SCN** | |
| Hidden channels per dimension | Same as input |
| Hidden layers | 2 |
| Readout | Sum of sums per dimension |
| **CELLMP** | |
| Hidden channels | 64 |
| Hidden layers | 10 |
| Dropout | 0.5 |
| Hidden dimension multiplier for final linear layer | 2 |
| Readout | Sum of sums per dimension |
| **CT** | |
| Hidden channels | 64 |
| Positional encoding type | Hodge Laplacian Eigenvectors |
| Positional encoding lengths | 8 |
| Hidden layers | 2 |
| Number of heads | 8 |
| Dropout | 0 |
| Hidden layers in the final MLP | 2 |
| Attention tensor diagram | Adjacent dimensions |
| Mask type | Sum |
| Readout | Average of dimension zero features |

## E ADDITIONAL EXPERIMENTAL DETAILS

Table 7 reports the mean and standard deviation of training iterations processed per second for each model and dataset. Sections E.1 to E.3 report the full set of experimental results.

**Feature vector initialization analysis** We observe different behaviours for the two families of models, graph-based and simplicial complex-based. For the graph models, random initialization works slightly better or equal than the degree features. On the other hand, for the simplicial complex models, upper- and lower-connectivity index initializations consistently outperform their random counterparts on average. Degrees and upper-connectivity indices for vertices coincide for both families of models, suggesting that higher-order connectivity indices contain more useful information than their dimension zero counterpart to predict topological properties, supporting the need for models that leverage higher-order information of the input. Having signal contained in features can make sense if the task in question requires additional information. For example, molecules are more than just combinatorial or topological objects: the types of atoms and the nature of bonds are important for predicting their properties. However, in purely topological tasks, such as predicting topological invariants, the need to enforce topological information into features raises the question: do MP-based models correctly capture topological properties in the first place? Still, standard deviations in the aggregated data for simplicial complex-based models is large, and better ablation is needed to fully understand the differences in initializations and the expressivity of higher-order indices in the context of topological prediction tasks.

Table 7: Mean and standard deviation of training iterations processed per second ($\uparrow$), as measured by PyTorch Lightning (Falcon & The PyTorch Lightning team, 2019), across all experiments for each model and dataset. The measurements are subject to variations caused by external server usage fluctuations.

| MODEL (CLASS) | $2\text{-}\mathcal{M}^0$ | $2\text{-}\mathcal{M}_H^0$ | $3\text{-}\mathcal{M}^0$ |
|---|---|---|---|
| MLP ($\mathcal{G}$) | $9.72 \pm 4.54$ | $12.39 \pm 13.22$ | $13.29 \pm 8.36$ |
| GAT ($\mathcal{G}$) | $9.41 \pm 4.02$ | $11.33 \pm 10.60$ | $11.01 \pm 6.52$ |
| UniMP ($\mathcal{G}$) | $9.31 \pm 3.79$ | $11.71 \pm 10.72$ | $12.42 \pm 6.80$ |
| TAG ($\mathcal{G}$) | $9.41 \pm 3.53$ | $11.46 \pm 10.66$ | $9.76 \pm 6.55$ |
| GCN ($\mathcal{G}$) | $9.45 \pm 3.79$ | $12.10 \pm 11.43$ | $12.72 \pm 7.59$ |
| SAN ($\mathcal{T}$) | $0.65 \pm 0.97$ | $1.21 \pm 2.93$ | $0.53 \pm 0.31$ |
| SCN ($\mathcal{T}$) | $0.83 \pm 2.63$ | $1.72 \pm 6.99$ | $0.83 \pm 0.64$ |
| SCCN ($\mathcal{T}$) | $0.85 \pm 3.06$ | $1.89 \pm 7.90$ | $0.80 \pm 0.53$ |
| SCCNN ($\mathcal{T}$) | $0.73 \pm 1.93$ | $1.67 \pm 5.79$ | $0.79 \pm 0.53$ |
| CellMP ($\mathcal{T}$) | $2.31 \pm 2.38$ | $2.32 \pm 2.43$ | $0.25 \pm 0.19$ |
| CT ($\mathcal{T}$) | $1.06 \pm 2.19$ | $1.16 \pm 2.77$ | $0.59 \pm 0.28$ |
| DECT ($\mathcal{T}$) | $6.59 \pm 3.63$ | $6.61 \pm 3.65$ | $12.78 \pm 8.03$ |

### E.1 BETTI NUMBER PREDICTION

Table 8: Full results for the Betti number prediction task on all datasets with mean and standard deviation reported over 5 runs. In this table, we report AUROC as performance metric. Transforms are abbreviated as DT (Degree Transform), DTO (Degree Transform Onehot) and RNF (Random Node Features).

| | | *AUROC* | | | | | | | | |
| | | $\beta_1$ | | | $\beta_2$ | | | $\beta_3$ | | |
| DATASET | MODEL (CLASS) | DT | DTO | RNF | DT | DTO | RNF | DT | DTO | RNF |
|---|---|---|---|---|---|---|---|---|---|---|
| $2\text{-}\mathcal{M}^0$ | GAT ($\mathcal{G}$) | $0.50 \pm 0.00$ | $\mathbf{0.50 \pm 0.00}$ | $0.50 \pm 0.00$ | $0.50 \pm 0.00$ | $\mathbf{0.50 \pm 0.00}$ | $0.50 \pm 0.00$ | | | |
| | GCN ($\mathcal{G}$) | $0.50 \pm 0.00$ | $\mathbf{0.50 \pm 0.00}$ | $0.50 \pm 0.00$ | $0.50 \pm 0.00$ | $\mathbf{0.50 \pm 0.00}$ | $0.50 \pm 0.00$ | | | |
| | MLP ($\mathcal{G}$) | $0.50 \pm 0.00$ | $\mathbf{0.50 \pm 0.00}$ | $0.50 \pm 0.00$ | $0.50 \pm 0.00$ | $\mathbf{0.50 \pm 0.00}$ | $0.50 \pm 0.00$ | | | |
| | TAG ($\mathcal{G}$) | $0.50 \pm 0.00$ | $\mathbf{0.50 \pm 0.00}$ | $0.50 \pm 0.00$ | $0.50 \pm 0.00$ | $\mathbf{0.50 \pm 0.00}$ | $0.50 \pm 0.00$ | | | |
| | UniMP ($\mathcal{G}$) | $0.50 \pm 0.00$ | $\mathbf{0.50 \pm 0.00}$ | $0.50 \pm 0.00$ | $0.50 \pm 0.00$ | $\mathbf{0.50 \pm 0.00}$ | $0.50 \pm 0.00$ | | | |
| | CellMP ($\mathcal{G}$) | $0.62 \pm 0.07$ | | $\mathbf{0.84 \pm 0.00}$ | $0.49 \pm 0.06$ | | $0.52 \pm 0.02$ | | | |
| | CT ($\mathcal{T}$) | $\mathbf{0.93 \pm 0.01}$ | | $0.66 \pm 0.02$ | $0.55 \pm 0.00$ | | $\mathbf{0.53 \pm 0.01}$ | | | |
| | DECT ($\mathcal{T}$) | $0.50 \pm 0.00$ | $0.50 \pm 0.00$ | $0.50 \pm 0.00$ | $0.50 \pm 0.00$ | $0.50 \pm 0.00$ | $0.50 \pm 0.00$ | | | |
| | SAN ($\mathcal{T}$) | $0.55 \pm 0.05$ | | $0.69 \pm 0.06$ | $0.52 \pm 0.21$ | | $\mathbf{0.53 \pm 0.01}$ | | | |
| | SCCN ($\mathcal{T}$) | $\mathbf{0.93 \pm 0.04}$ | | $0.78 \pm 0.04$ | $0.55 \pm 0.00$ | | $\mathbf{0.53 \pm 0.01}$ | | | |
| | SCCNN ($\mathcal{T}$) | $0.50 \pm 0.01$ | | $0.50 \pm 0.02$ | $0.50 \pm 0.19$ | | $0.52 \pm 0.04$ | | | |
| | SCN ($\mathcal{T}$) | $0.56 \pm 0.13$ | | $0.51 \pm 0.03$ | $\mathbf{0.63 \pm 0.17}$ | | $0.48 \pm 0.07$ | | | |
| $3\text{-}\mathcal{M}^0$ | GAT ($\mathcal{G}$) | $\mathbf{0.23 \pm 0.00}$ | $\mathbf{0.23 \pm 0.00}$ | $0.23 \pm 0.00$ | $0.12 \pm 0.00$ | $\mathbf{0.12 \pm 0.00}$ | $\mathbf{0.12 \pm 0.00}$ | $0.14 \pm 0.00$ | $\mathbf{0.14 \pm 0.00}$ | $0.14 \pm 0.00$ |
| | GCN ($\mathcal{G}$) | $\mathbf{0.23 \pm 0.00}$ | $\mathbf{0.23 \pm 0.00}$ | $0.23 \pm 0.00$ | $0.12 \pm 0.00$ | $\mathbf{0.12 \pm 0.00}$ | $\mathbf{0.12 \pm 0.00}$ | $0.14 \pm 0.00$ | $\mathbf{0.14 \pm 0.00}$ | $0.14 \pm 0.00$ |
| | MLP ($\mathcal{G}$) | $\mathbf{0.23 \pm 0.00}$ | $\mathbf{0.23 \pm 0.00}$ | $0.23 \pm 0.00$ | $0.12 \pm 0.00$ | $\mathbf{0.12 \pm 0.00}$ | $\mathbf{0.12 \pm 0.00}$ | $0.14 \pm 0.00$ | $\mathbf{0.14 \pm 0.00}$ | $0.14 \pm 0.00$ |
| | TAG ($\mathcal{G}$) | $\mathbf{0.23 \pm 0.00}$ | $\mathbf{0.23 \pm 0.00}$ | $0.23 \pm 0.00$ | $0.12 \pm 0.00$ | $\mathbf{0.12 \pm 0.00}$ | $\mathbf{0.12 \pm 0.00}$ | $0.14 \pm 0.00$ | $\mathbf{0.14 \pm 0.00}$ | $0.14 \pm 0.00$ |
| | UniMP ($\mathcal{G}$) | $\mathbf{0.23 \pm 0.00}$ | $\mathbf{0.23 \pm 0.00}$ | $0.23 \pm 0.00$ | $0.12 \pm 0.00$ | $\mathbf{0.12 \pm 0.00}$ | $\mathbf{0.12 \pm 0.00}$ | $0.14 \pm 0.00$ | $\mathbf{0.14 \pm 0.00}$ | $0.14 \pm 0.00$ |
| | CellMP ($\mathcal{G}$) | $\mathbf{0.23 \pm 0.00}$ | | $0.23 \pm 0.00$ | $0.12 \pm 0.00$ | | $\mathbf{0.12 \pm 0.00}$ | $0.14 \pm 0.00$ | | $0.14 \pm 0.00$ |
| | CT ($\mathcal{T}$) | $\mathbf{0.23 \pm 0.00}$ | | $0.23 \pm 0.00$ | $0.12 \pm 0.00$ | | $\mathbf{0.12 \pm 0.00}$ | $0.14 \pm 0.00$ | | $0.14 \pm 0.00$ |
| | DECT ($\mathcal{T}$) | $\mathbf{0.23 \pm 0.00}$ | $0.23 \pm 0.00$ | $0.23 \pm 0.00$ | $0.12 \pm 0.00$ | $\mathbf{0.12 \pm 0.00}$ | $\mathbf{0.12 \pm 0.00}$ | $0.14 \pm 0.00$ | $\mathbf{0.14 \pm 0.00}$ | $0.14 \pm 0.00$ |
| | SAN ($\mathcal{T}$) | $0.17 \pm 0.09$ | | $\mathbf{0.24 \pm 0.01}$ | $0.12 \pm 0.05$ | | $\mathbf{0.12 \pm 0.00}$ | $\mathbf{0.19 \pm 0.04}$ | | $\mathbf{0.15 \pm 0.01}$ |
| | SCCN ($\mathcal{T}$) | $\mathbf{0.23 \pm 0.00}$ | | $0.23 \pm 0.00$ | $0.12 \pm 0.00$ | | $\mathbf{0.12 \pm 0.00}$ | $0.14 \pm 0.00$ | | $0.14 \pm 0.00$ |
| | SCCNN ($\mathcal{T}$) | $0.21 \pm 0.11$ | | $0.20 \pm 0.05$ | $0.12 \pm 0.04$ | | $0.11 \pm 0.01$ | $0.11 \pm 0.05$ | | $0.13 \pm 0.02$ |
| | SCN ($\mathcal{T}$) | $0.20 \pm 0.04$ | | $0.23 \pm 0.00$ | $\mathbf{0.15 \pm 0.04}$ | | $\mathbf{0.12 \pm 0.00}$ | $0.11 \pm 0.07$ | | $0.14 \pm 0.02$ |
| $2\text{-}\mathcal{M}_H^0$ | GAT ($\mathcal{G}$) | $0.21 \pm 0.00$ | $\mathbf{0.21 \pm 0.00}$ | $0.21 \pm 0.00$ | $0.50 \pm 0.00$ | $\mathbf{0.50 \pm 0.00}$ | $0.50 \pm 0.00$ | | | |
| | GCN ($\mathcal{G}$) | $0.21 \pm 0.00$ | $\mathbf{0.21 \pm 0.00}$ | $0.21 \pm 0.00$ | $0.50 \pm 0.00$ | $\mathbf{0.50 \pm 0.00}$ | $0.50 \pm 0.00$ | | | |
| | MLP ($\mathcal{G}$) | $0.21 \pm 0.00$ | $\mathbf{0.21 \pm 0.00}$ | $0.21 \pm 0.00$ | $0.50 \pm 0.00$ | $\mathbf{0.50 \pm 0.00}$ | $0.50 \pm 0.00$ | | | |
| | TAG ($\mathcal{G}$) | $0.21 \pm 0.00$ | $\mathbf{0.21 \pm 0.00}$ | $0.21 \pm 0.00$ | $0.50 \pm 0.00$ | $\mathbf{0.50 \pm 0.00}$ | $0.50 \pm 0.00$ | | | |
| | UniMP ($\mathcal{G}$) | $0.21 \pm 0.00$ | $\mathbf{0.21 \pm 0.00}$ | $0.21 \pm 0.00$ | $0.50 \pm 0.00$ | $\mathbf{0.50 \pm 0.00}$ | $0.50 \pm 0.00$ | | | |
| | CellMP ($\mathcal{G}$) | $0.23 \pm 0.01$ | | $\mathbf{0.29 \pm 0.01}$ | $0.52 \pm 0.04$ | | $0.51 \pm 0.02$ | | | |
| | CT ($\mathcal{T}$) | $0.27 \pm 0.01$ | | $0.21 \pm 0.00$ | $\mathbf{0.52 \pm 0.03}$ | | $0.50 \pm 0.00$ | | | |
| | DECT ($\mathcal{T}$) | $0.21 \pm 0.00$ | $0.21 \pm 0.00$ | $0.21 \pm 0.00$ | $0.50 \pm 0.00$ | $0.50 \pm 0.00$ | $0.50 \pm 0.00$ | | | |
| | SAN ($\mathcal{T}$) | $0.25 \pm 0.01$ | | $0.22 \pm 0.02$ | $0.48 \pm 0.04$ | | $0.50 \pm 0.02$ | | | |
| | SCCN ($\mathcal{T}$) | $\mathbf{0.29 \pm 0.01}$ | | $0.23 \pm 0.01$ | $\mathbf{0.52 \pm 0.01}$ | | $0.50 \pm 0.02$ | | | |
| | SCCNN ($\mathcal{T}$) | $0.20 \pm 0.05$ | | $0.23 \pm 0.03$ | $0.49 \pm 0.03$ | | $\mathbf{0.51 \pm 0.02}$ | | | |
| | SCN ($\mathcal{T}$) | $0.22 \pm 0.00$ | | $0.21 \pm 0.00$ | $0.49 \pm 0.02$ | | $0.50 \pm 0.01$ | | | |
| $2\text{-}\mathcal{M}_H^1$ | GAT ($\mathcal{G}$) | $0.22 \pm 0.00$ | $0.21 \pm 0.00$ | $0.21 \pm 0.00$ | $0.50 \pm 0.00$ | $\mathbf{0.50 \pm 0.00}$ | $0.50 \pm 0.00$ | | | |
| | GCN ($\mathcal{G}$) | $0.22 \pm 0.00$ | $0.21 \pm 0.00$ | $0.21 \pm 0.00$ | $0.50 \pm 0.00$ | $\mathbf{0.50 \pm 0.00}$ | $0.50 \pm 0.00$ | | | |
| | MLP ($\mathcal{G}$) | $0.21 \pm 0.01$ | $0.21 \pm 0.00$ | $0.21 \pm 0.00$ | $0.50 \pm 0.00$ | $\mathbf{0.50 \pm 0.00}$ | $0.50 \pm 0.00$ | | | |
| | TAG ($\mathcal{G}$) | $0.22 \pm 0.00$ | $\mathbf{0.22 \pm 0.00}$ | $0.22 \pm 0.00$ | $0.50 \pm 0.00$ | $\mathbf{0.50 \pm 0.00}$ | $0.50 \pm 0.00$ | | | |
| | UniMP ($\mathcal{G}$) | $0.22 \pm 0.00$ | $\mathbf{0.22 \pm 0.00}$ | $0.22 \pm 0.00$ | $0.50 \pm 0.00$ | $\mathbf{0.50 \pm 0.00}$ | $0.50 \pm 0.00$ | | | |
| | CellMP ($\mathcal{T}$) | $0.23 \pm 0.03$ | | $\mathbf{0.26 \pm 0.00}$ | $0.49 \pm 0.01$ | | $0.49 \pm 0.01$ | | | |
| | CT ($\mathcal{T}$) | $0.23 \pm 0.02$ | | $0.21 \pm 0.00$ | $0.50 \pm 0.00$ | | $0.50 \pm 0.00$ | | | |
| | SAN ($\mathcal{T}$) | $0.24 \pm 0.01$ | | $0.22 \pm 0.01$ | $0.50 \pm 0.00$ | | $0.49 \pm 0.01$ | | | |
| | SCCN ($\mathcal{T}$) | $\mathbf{0.27 \pm 0.01}$ | | $0.22 \pm 0.01$ | $\mathbf{0.52 \pm 0.02}$ | | $\mathbf{0.51 \pm 0.01}$ | | | |
| | SCCNN ($\mathcal{T}$) | $0.21 \pm 0.03$ | | $0.22 \pm 0.02$ | $0.50 \pm 0.01$ | | $0.50 \pm 0.01$ | | | |
| | SCN ($\mathcal{T}$) | $0.21 \pm 0.00$ | | $0.21 \pm 0.00$ | $0.49 \pm 0.02$ | | $\mathbf{0.51 \pm 0.01}$ | | | |

Table 9: Full results for the Betti number prediction task on all datasets with mean and standard deviation reported over 5 runs. In this table, we report accuracy as performance metric. Transforms are abbreviated as DT (Degree Transform), DTO (Degree Transform Onehot) and RNF (Random Node Features).

| | | | $\beta_0$ | | | $\beta_1$ | | | $\beta_2$ | | | $\beta_3$ | |
|---|---|---|---|---|---|---|---|---|---|---|---|---|---|
| Dataset | Model (Class) | DT | DTO | RNF | DT | DTO | RNF | DT | DTO | RNF | DT | DTO | RNF |
| $2\text{-}\mathcal{M}^0$ | GAT ($\mathcal{G}$) | **1.00±0.00** | **1.00±0.00** | **1.00±0.00** | 0.31±0.00 | 0.31±0.00 | 0.31±0.00 | 0.92±0.00 | **0.92±0.00** | **0.92±0.00** | | | |
| | GCN ($\mathcal{G}$) | **1.00±0.00** | **1.00±0.00** | **1.00±0.00** | 0.31±0.00 | 0.31±0.00 | 0.31±0.00 | 0.92±0.00 | **0.92±0.00** | **0.92±0.00** | | | |
| | MLP ($\mathcal{G}$) | **1.00±0.00** | **1.00±0.00** | **1.00±0.00** | 0.31±0.00 | 0.31±0.00 | 0.31±0.00 | 0.92±0.00 | **0.92±0.00** | **0.92±0.00** | | | |
| | TAG ($\mathcal{G}$) | **1.00±0.00** | **1.00±0.00** | **1.00±0.00** | 0.32±0.01 | **0.33±0.01** | 0.32±0.00 | 0.92±0.00 | **0.92±0.00** | **0.92±0.00** | | | |
| | UniMP ($\mathcal{G}$) | **1.00±0.00** | **1.00±0.00** | **1.00±0.00** | 0.33±0.00 | 0.32±0.01 | 0.32±0.01 | 0.92±0.00 | **0.92±0.00** | **0.92±0.00** | | | |
| | CellMP ($\mathcal{G}$) | 0.46±0.50 | | **1.00±0.00** | 0.39±0.35 | | **0.90±0.01** | 0.46±0.44 | | **0.92±0.00** | | | |
| | CT ($\mathcal{T}$) | **1.00±0.00** | | **1.00±0.00** | **0.93±0.00** | | 0.87±0.00 | **0.93±0.00** | | **0.92±0.00** | | | |
| | DECT ($\mathcal{T}$) | **1.00±0.00** | **1.00±0.00** | **1.00±0.00** | 0.32±0.00 | 0.32±0.00 | 0.32±0.00 | 0.92±0.00 | **0.92±0.00** | **0.92±0.00** | | | |
| | SAN ($\mathcal{T}$) | 0.09±0.04 | | 0.57±0.18 | 0.12±0.10 | | 0.54±0.11 | 0.52±0.14 | | 0.73±0.08 | | | |
| | SCCN ($\mathcal{T}$) | **1.00±0.00** | | 0.71±0.06 | **0.93±0.00** | | 0.67±0.05 | **0.93±0.00** | | 0.79±0.04 | | | |
| | SCCNN ($\mathcal{T}$) | 0.00±0.00 | | 0.01±0.00 | 0.03±0.02 | | 0.03±0.01 | 0.33±0.37 | | 0.49±0.12 | | | |
| | SCN ($\mathcal{T}$) | 0.33±0.38 | | 0.29±0.07 | 0.21±0.26 | | 0.25±0.10 | 0.62±0.36 | | 0.65±0.08 | | | |
| $3\text{-}\mathcal{M}^0$ | GAT ($\mathcal{G}$) | **1.00±0.00** | **1.00±0.00** | **1.00±0.00** | **1.00±0.00** | **1.00±0.00** | **1.00±0.00** | **1.00±0.00** | **1.00±0.00** | **1.00±0.00** | **1.00±0.00** | **1.00±0.00** | **1.00±0.00** |
| | GCN ($\mathcal{G}$) | **1.00±0.00** | **1.00±0.00** | **1.00±0.00** | **1.00±0.00** | **1.00±0.00** | **1.00±0.00** | **1.00±0.00** | **1.00±0.00** | **1.00±0.00** | **1.00±0.00** | **1.00±0.00** | **1.00±0.00** |
| | MLP ($\mathcal{G}$) | **1.00±0.00** | **1.00±0.00** | **1.00±0.00** | **1.00±0.00** | **1.00±0.00** | **1.00±0.00** | **1.00±0.00** | **1.00±0.00** | **1.00±0.00** | **1.00±0.00** | **1.00±0.00** | **1.00±0.00** |
| | TAG ($\mathcal{G}$) | **1.00±0.00** | **1.00±0.00** | **1.00±0.00** | **1.00±0.00** | **1.00±0.00** | **1.00±0.00** | **1.00±0.00** | **1.00±0.00** | **1.00±0.00** | **1.00±0.00** | **1.00±0.00** | **1.00±0.00** |
| | UniMP ($\mathcal{G}$) | **1.00±0.00** | **1.00±0.00** | **1.00±0.00** | **1.00±0.00** | **1.00±0.00** | **1.00±0.00** | **1.00±0.00** | **1.00±0.00** | **1.00±0.00** | **1.00±0.00** | **1.00±0.00** | **1.00±0.00** |
| | CellMP ($\mathcal{G}$) | **1.00±0.00** | | **1.00±0.00** | **1.00±0.00** | | **1.00±0.00** | **1.00±0.00** | | **1.00±0.00** | **1.00±0.00** | | **1.00±0.00** |
| | CT ($\mathcal{T}$) | **1.00±0.00** | | **1.00±0.00** | **1.00±0.00** | | **1.00±0.00** | **1.00±0.00** | | **1.00±0.00** | **1.00±0.00** | | **1.00±0.00** |
| | DECT ($\mathcal{T}$) | **1.00±0.00** | **1.00±0.00** | **1.00±0.00** | **1.00±0.00** | **1.00±0.00** | **1.00±0.00** | **1.00±0.00** | **1.00±0.00** | **1.00±0.00** | **1.00±0.00** | **1.00±0.00** | **1.00±0.00** |
| | SAN ($\mathcal{T}$) | 0.01±0.00 | | 0.51±0.12 | 0.49±0.13 | | 0.71±0.11 | 0.51±0.22 | | 0.78±0.10 | 0.01±0.01 | | 0.52±0.07 |
| | SCCN ($\mathcal{T}$) | **1.00±0.00** | | **1.00±0.00** | **1.00±0.00** | | **1.00±0.00** | **1.00±0.00** | | **1.00±0.00** | **1.00±0.00** | | **1.00±0.00** |
| | SCCNN ($\mathcal{T}$) | 0.00±0.00 | | 0.00±0.00 | 0.48±0.14 | | 0.48±0.05 | 0.60±0.08 | | 0.49±0.12 | 0.00±0.00 | | 0.00±0.00 |
| | SCN ($\mathcal{T}$) | 0.95±0.06 | | 0.95±0.08 | 0.85±0.19 | | 0.99±0.01 | 0.80±0.16 | | 0.99±0.00 | 0.58±0.31 | | 0.92±0.08 |
| $2\text{-}\mathcal{M}_H^0$ | GAT ($\mathcal{G}$) | **1.00±0.00** | **1.00±0.00** | **1.00±0.00** | 0.54±0.00 | **0.54±0.00** | 0.54±0.00 | 0.70±0.00 | **0.70±0.00** | **0.70±0.00** | | | |
| | GCN ($\mathcal{G}$) | **1.00±0.00** | **1.00±0.00** | **1.00±0.00** | 0.54±0.00 | **0.54±0.00** | 0.54±0.00 | 0.70±0.00 | **0.70±0.00** | **0.70±0.00** | | | |
| | MLP ($\mathcal{G}$) | **1.00±0.00** | **1.00±0.00** | **1.00±0.00** | 0.54±0.00 | **0.54±0.00** | 0.54±0.00 | 0.70±0.00 | **0.70±0.00** | **0.70±0.00** | | | |
| | TAG ($\mathcal{G}$) | **1.00±0.00** | **1.00±0.00** | **1.00±0.00** | 0.54±0.00 | **0.54±0.00** | 0.54±0.00 | 0.70±0.00 | **0.70±0.00** | **0.70±0.00** | | | |
| | UniMP ($\mathcal{G}$) | **1.00±0.00** | **1.00±0.00** | **1.00±0.00** | 0.54±0.00 | **0.54±0.00** | 0.54±0.00 | 0.70±0.00 | **0.70±0.00** | **0.70±0.00** | | | |
| | CellMP ($\mathcal{G}$) | 0.05±0.09 | | 0.98±0.00 | 0.18±0.24 | | **0.65±0.01** | 0.14±0.31 | | 0.69±0.01 | | | |
| | CT ($\mathcal{T}$) | **1.00±0.00** | | **1.00±0.00** | 0.36±0.16 | | 0.54±0.00 | 0.64±0.17 | | **0.70±0.01** | | | |
| | DECT ($\mathcal{T}$) | **1.00±0.00** | **1.00±0.00** | **1.00±0.00** | 0.54±0.00 | **0.54±0.00** | 0.54±0.00 | 0.70±0.00 | **0.70±0.00** | **0.70±0.00** | | | |
| | SAN ($\mathcal{T}$) | 0.07±0.06 | | 0.26±0.16 | 0.26±0.05 | | 0.26±0.11 | 0.43±0.09 | | 0.43±0.06 | | | |
| | SCCN ($\mathcal{T}$) | **1.00±0.00** | | 0.48±0.03 | **0.69±0.03** | | 0.40±0.03 | **0.71±0.01** | | 0.52±0.01 | | | |
| | SCCNN ($\mathcal{T}$) | 0.00±0.00 | | 0.01±0.00 | 0.08±0.10 | | 0.12±0.08 | 0.27±0.29 | | 0.35±0.08 | | | |
| | SCN ($\mathcal{T}$) | 0.01±0.02 | | 0.13±0.03 | 0.20±0.01 | | 0.19±0.03 | 0.25±0.35 | | 0.43±0.03 | | | |
| $2\text{-}\mathcal{M}_H^1$ | GAT ($\mathcal{G}$) | 0.00±0.00 | 0.64±0.50 | **1.00±0.00** | 0.20±0.00 | 0.43±0.16 | **0.54±0.00** | **0.70±0.00** | **0.70±0.00** | **0.70±0.00** | | | |
| | GCN ($\mathcal{G}$) | 0.00±0.00 | 0.68±0.46 | **1.00±0.00** | 0.20±0.00 | 0.47±0.15 | **0.54±0.00** | **0.70±0.00** | **0.70±0.00** | **0.70±0.00** | | | |
| | MLP ($\mathcal{G}$) | 0.53±0.49 | **1.00±0.00** | **1.00±0.00** | 0.34±0.14 | **0.54±0.00** | **0.54±0.00** | **0.70±0.00** | **0.70±0.00** | **0.70±0.00** | | | |
| | TAG ($\mathcal{G}$) | 0.00±0.00 | 0.00±0.00 | 0.01±0.01 | 0.20±0.00 | 0.20±0.00 | 0.20±0.00 | **0.70±0.00** | **0.70±0.00** | **0.70±0.00** | | | |
| | UniMP ($\mathcal{G}$) | 0.00±0.00 | 0.01±0.01 | 0.02±0.01 | 0.20±0.00 | 0.20±0.00 | 0.20±0.01 | **0.70±0.00** | **0.70±0.00** | **0.70±0.00** | | | |
| | CellMP ($\mathcal{T}$) | 0.00±0.00 | | 0.00±0.00 | 0.03±0.06 | | 0.04±0.02 | 0.09±0.21 | | 0.30±0.00 | | | |
| | CT ($\mathcal{T}$) | **1.00±0.00** | | **1.00±0.00** | **0.54±0.00** | | **0.54±0.00** | 0.62±0.18 | | **0.70±0.00** | | | |
| | SAN ($\mathcal{T}$) | 0.00±0.00 | | 0.01±0.01 | 0.00±0.00 | | 0.09±0.06 | 0.56±0.31 | | 0.41±0.31 | | | |
| | SCCN ($\mathcal{T}$) | 0.07±0.15 | | 0.03±0.03 | 0.05±0.12 | | 0.02±0.01 | 0.10±0.13 | | 0.25±0.10 | | | |
| | SCCNN ($\mathcal{T}$) | 0.00±0.00 | | 0.00±0.00 | 0.12±0.11 | | 0.07±0.09 | 0.48±0.32 | | 0.24±0.33 | | | |
| | SCN ($\mathcal{T}$) | 0.00±0.00 | | 0.03±0.01 | 0.16±0.09 | | 0.13±0.05 | 0.28±0.39 | | 0.35±0.16 | | | |

### E.2 ORIENTABILITY PREDICTION

Table 10: Full results for the orientability prediction task on all datasets with AUROC and accuracy reported. Mean and standard deviation are taken over 5 runs. Transforms are abbreviated as DT (Degree Transform), DTO (Degree Transform Onehot) and RNF (Random Node Features).

| DATASET | MODEL (CLASS) | AUROC DT | AUROC DTO | AUROC RNF | Accuracy DT | Accuracy DTO | Accuracy RNF |
|---|---|---|---|---|---|---|---|
| $2\text{-}\mathcal{M}_0$ | GAT ($\mathcal{G}$) | $0.50 \pm 0.00$ | $\mathbf{0.50 \pm 0.00}$ | $0.50 \pm 0.00$ | $0.92 \pm 0.00$ | $\mathbf{0.92 \pm 0.00}$ | $0.92 \pm 0.00$ |
| | GCN ($\mathcal{G}$) | $0.50 \pm 0.00$ | $\mathbf{0.50 \pm 0.00}$ | $0.50 \pm 0.00$ | $0.92 \pm 0.00$ | $\mathbf{0.92 \pm 0.00}$ | $0.92 \pm 0.00$ |
| | MLP ($\mathcal{G}$) | $0.50 \pm 0.00$ | $\mathbf{0.50 \pm 0.00}$ | $0.50 \pm 0.00$ | $0.92 \pm 0.00$ | $\mathbf{0.92 \pm 0.00}$ | $0.92 \pm 0.00$ |
| | TAG ($\mathcal{G}$) | $0.50 \pm 0.00$ | $\mathbf{0.50 \pm 0.00}$ | $0.50 \pm 0.00$ | $0.92 \pm 0.00$ | $\mathbf{0.92 \pm 0.00}$ | $0.92 \pm 0.00$ |
| | UniMP ($\mathcal{G}$) | $0.50 \pm 0.00$ | $\mathbf{0.50 \pm 0.00}$ | $0.50 \pm 0.00$ | $0.92 \pm 0.00$ | $\mathbf{0.92 \pm 0.00}$ | $0.92 \pm 0.00$ |
| | CellMP ($\mathcal{T}$) | $\mathbf{0.65 \pm 0.07}$ | | $\mathbf{0.55 \pm 0.00}$ | $0.64 \pm 0.26$ | | $\mathbf{0.93 \pm 0.00}$ |
| | CT ($\mathcal{T}$) | $0.55 \pm 0.00$ | | $0.50 \pm 0.00$ | $\mathbf{0.93 \pm 0.00}$ | | $0.92 \pm 0.00$ |
| | DECT ($\mathcal{T}$) | $0.50 \pm 0.00$ | $0.50 \pm 0.00$ | $0.50 \pm 0.00$ | $0.92 \pm 0.00$ | $\mathbf{0.92 \pm 0.00}$ | $0.92 \pm 0.00$ |
| | SAN ($\mathcal{T}$) | $0.52 \pm 0.03$ | | $0.51 \pm 0.02$ | $0.92 \pm 0.00$ | | $0.92 \pm 0.01$ |
| | SCCN ($\mathcal{T}$) | $0.55 \pm 0.01$ | | $0.54 \pm 0.01$ | $\mathbf{0.93 \pm 0.00}$ | | $\mathbf{0.93 \pm 0.00}$ |
| | SCCNN ($\mathcal{T}$) | $0.55 \pm 0.09$ | | $0.53 \pm 0.02$ | $0.87 \pm 0.11$ | | $0.91 \pm 0.01$ |
| | SCN ($\mathcal{T}$) | $0.53 \pm 0.04$ | | $0.50 \pm 0.01$ | $0.91 \pm 0.02$ | | $0.92 \pm 0.01$ |
| $3\text{-}\mathcal{M}_0$ | GAT ($\mathcal{G}$) | $0.14 \pm 0.00$ | $\mathbf{0.14 \pm 0.00}$ | $\mathbf{0.14 \pm 0.00}$ | $\mathbf{1.00 \pm 0.00}$ | $\mathbf{1.00 \pm 0.00}$ | $\mathbf{1.00 \pm 0.00}$ |
| | GCN ($\mathcal{G}$) | $0.14 \pm 0.00$ | $\mathbf{0.14 \pm 0.00}$ | $\mathbf{0.14 \pm 0.00}$ | $\mathbf{1.00 \pm 0.00}$ | $\mathbf{1.00 \pm 0.00}$ | $\mathbf{1.00 \pm 0.00}$ |
| | MLP ($\mathcal{G}$) | $0.14 \pm 0.00$ | $\mathbf{0.14 \pm 0.00}$ | $\mathbf{0.14 \pm 0.00}$ | $\mathbf{1.00 \pm 0.00}$ | $\mathbf{1.00 \pm 0.00}$ | $\mathbf{1.00 \pm 0.00}$ |
| | TAG ($\mathcal{G}$) | $0.14 \pm 0.00$ | $\mathbf{0.14 \pm 0.00}$ | $\mathbf{0.14 \pm 0.00}$ | $\mathbf{1.00 \pm 0.00}$ | $\mathbf{1.00 \pm 0.00}$ | $\mathbf{1.00 \pm 0.00}$ |
| | UniMP ($\mathcal{G}$) | $0.14 \pm 0.00$ | $\mathbf{0.14 \pm 0.00}$ | $\mathbf{0.14 \pm 0.00}$ | $\mathbf{1.00 \pm 0.00}$ | $\mathbf{1.00 \pm 0.00}$ | $\mathbf{1.00 \pm 0.00}$ |
| | CellMP ($\mathcal{T}$) | $\mathbf{0.18 \pm 0.04}$ | | $\mathbf{0.14 \pm 0.00}$ | $\mathbf{1.00 \pm 0.00}$ | | $\mathbf{1.00 \pm 0.00}$ |
| | CT ($\mathcal{T}$) | $0.14 \pm 0.00$ | | $\mathbf{0.14 \pm 0.00}$ | $\mathbf{1.00 \pm 0.00}$ | | $\mathbf{1.00 \pm 0.00}$ |
| | DECT ($\mathcal{T}$) | $0.14 \pm 0.00$ | $\mathbf{0.14 \pm 0.00}$ | $\mathbf{0.14 \pm 0.00}$ | $\mathbf{1.00 \pm 0.00}$ | $\mathbf{1.00 \pm 0.00}$ | $\mathbf{1.00 \pm 0.00}$ |
| | SAN ($\mathcal{T}$) | $0.14 \pm 0.00$ | | $\mathbf{0.14 \pm 0.00}$ | $\mathbf{1.00 \pm 0.00}$ | | $\mathbf{1.00 \pm 0.00}$ |
| | SCCN ($\mathcal{T}$) | $0.14 \pm 0.00$ | | $\mathbf{0.14 \pm 0.00}$ | $\mathbf{1.00 \pm 0.00}$ | | $\mathbf{1.00 \pm 0.00}$ |
| | SCCNN ($\mathcal{T}$) | $0.14 \pm 0.00$ | | $\mathbf{0.14 \pm 0.00}$ | $\mathbf{1.00 \pm 0.00}$ | | $\mathbf{1.00 \pm 0.00}$ |
| | SCN ($\mathcal{T}$) | $0.14 \pm 0.00$ | | $\mathbf{0.14 \pm 0.00}$ | $\mathbf{1.00 \pm 0.00}$ | | $\mathbf{1.00 \pm 0.00}$ |
| $2\text{-}\mathcal{M}_H^0$ | GAT ($\mathcal{G}$) | $0.50 \pm 0.00$ | $\mathbf{0.50 \pm 0.00}$ | $0.50 \pm 0.00$ | $0.70 \pm 0.00$ | $\mathbf{0.70 \pm 0.00}$ | $\mathbf{0.70 \pm 0.00}$ |
| | GCN ($\mathcal{G}$) | $0.50 \pm 0.00$ | $\mathbf{0.50 \pm 0.00}$ | $0.50 \pm 0.00$ | $0.70 \pm 0.00$ | $\mathbf{0.70 \pm 0.00}$ | $\mathbf{0.70 \pm 0.00}$ |
| | MLP ($\mathcal{G}$) | $0.50 \pm 0.00$ | $\mathbf{0.50 \pm 0.00}$ | $0.50 \pm 0.00$ | $0.70 \pm 0.00$ | $\mathbf{0.70 \pm 0.00}$ | $\mathbf{0.70 \pm 0.00}$ |
| | TAG ($\mathcal{G}$) | $0.50 \pm 0.00$ | $\mathbf{0.50 \pm 0.00}$ | $0.50 \pm 0.00$ | $0.70 \pm 0.00$ | $\mathbf{0.70 \pm 0.00}$ | $\mathbf{0.70 \pm 0.00}$ |
| | UniMP ($\mathcal{G}$) | $0.50 \pm 0.00$ | $\mathbf{0.50 \pm 0.00}$ | $0.50 \pm 0.00$ | $0.70 \pm 0.00$ | $\mathbf{0.70 \pm 0.00}$ | $\mathbf{0.70 \pm 0.00}$ |
| | CellMP ($\mathcal{T}$) | $0.51 \pm 0.01$ | | $0.50 \pm 0.01$ | $0.31 \pm 0.02$ | | $\mathbf{0.70 \pm 0.01}$ |
| | CT ($\mathcal{T}$) | $0.52 \pm 0.03$ | | $0.50 \pm 0.00$ | $0.72 \pm 0.02$ | | $\mathbf{0.70 \pm 0.00}$ |
| | DECT ($\mathcal{T}$) | $0.50 \pm 0.00$ | $\mathbf{0.50 \pm 0.00}$ | $0.50 \pm 0.00$ | $0.70 \pm 0.00$ | $\mathbf{0.70 \pm 0.00}$ | $\mathbf{0.70 \pm 0.00}$ |
| | SAN ($\mathcal{T}$) | $0.50 \pm 0.02$ | | $\mathbf{0.51 \pm 0.02}$ | $0.59 \pm 0.08$ | | $0.60 \pm 0.03$ |
| | SCCN ($\mathcal{T}$) | $\mathbf{0.54 \pm 0.01}$ | | $0.50 \pm 0.01$ | $\mathbf{0.73 \pm 0.00}$ | | $0.65 \pm 0.02$ |
| | SCCNN ($\mathcal{T}$) | $0.50 \pm 0.01$ | | $0.50 \pm 0.01$ | $0.54 \pm 0.23$ | | $0.59 \pm 0.10$ |
| | SCN ($\mathcal{T}$) | $0.51 \pm 0.02$ | | $\mathbf{0.51 \pm 0.01}$ | $0.55 \pm 0.23$ | | $0.61 \pm 0.01$ |
| $2\text{-}\mathcal{M}_H^1$ | GAT ($\mathcal{G}$) | $\mathbf{0.50 \pm 0.00}$ | $\mathbf{0.50 \pm 0.00}$ | $0.50 \pm 0.00$ | $\mathbf{0.70 \pm 0.00}$ | $\mathbf{0.70 \pm 0.00}$ | $\mathbf{0.70 \pm 0.00}$ |
| | GCN ($\mathcal{G}$) | $\mathbf{0.50 \pm 0.00}$ | $\mathbf{0.50 \pm 0.00}$ | $0.50 \pm 0.00$ | $\mathbf{0.70 \pm 0.00}$ | $\mathbf{0.70 \pm 0.00}$ | $\mathbf{0.70 \pm 0.00}$ |
| | MLP ($\mathcal{G}$) | $\mathbf{0.50 \pm 0.00}$ | $\mathbf{0.50 \pm 0.00}$ | $0.50 \pm 0.00$ | $\mathbf{0.70 \pm 0.00}$ | $\mathbf{0.70 \pm 0.00}$ | $\mathbf{0.70 \pm 0.00}$ |
| | TAG ($\mathcal{G}$) | $\mathbf{0.50 \pm 0.00}$ | $\mathbf{0.50 \pm 0.01}$ | $0.50 \pm 0.00$ | $0.64 \pm 0.15$ | $0.65 \pm 0.10$ | $\mathbf{0.70 \pm 0.00}$ |
| | UniMP ($\mathcal{G}$) | $\mathbf{0.50 \pm 0.01}$ | $\mathbf{0.50 \pm 0.00}$ | $0.50 \pm 0.00$ | $0.60 \pm 0.15$ | $\mathbf{0.70 \pm 0.00}$ | $\mathbf{0.70 \pm 0.00}$ |
| | CellMP ($\mathcal{T}$) | $\mathbf{0.50 \pm 0.01}$ | | $0.50 \pm 0.00$ | $0.38 \pm 0.19$ | | $\mathbf{0.70 \pm 0.00}$ |
| | CT ($\mathcal{T}$) | $\mathbf{0.50 \pm 0.00}$ | | $0.50 \pm 0.00$ | $\mathbf{0.70 \pm 0.00}$ | | $\mathbf{0.70 \pm 0.00}$ |
| | SAN ($\mathcal{T}$) | $\mathbf{0.50 \pm 0.00}$ | | $0.50 \pm 0.01$ | $0.54 \pm 0.22$ | | $0.49 \pm 0.18$ |
| | SCCN ($\mathcal{T}$) | $\mathbf{0.50 \pm 0.00}$ | | $\mathbf{0.51 \pm 0.01}$ | $\mathbf{0.70 \pm 0.00}$ | | $0.69 \pm 0.02$ |
| | SCCNN ($\mathcal{T}$) | $\mathbf{0.50 \pm 0.00}$ | | $0.50 \pm 0.01$ | $0.46 \pm 0.22$ | | $0.55 \pm 0.17$ |
| | SCN ($\mathcal{T}$) | $\mathbf{0.50 \pm 0.00}$ | | $0.50 \pm 0.00$ | $0.54 \pm 0.22$ | | $0.68 \pm 0.04$ |

### E.3 HOMEOMORPHISM PREDICTION

Table 11: Full results for the homeomorphism type prediction task on the full set of surfaces. Performances are reported with mean and standard deviation over five runs with different seeds.

| DATASET | MODEL (CLASS) | AUROC | | | Accuracy | | |
|---|---|---|---|---|---|---|---|
| | | DT | DTO | RNF | DT | DTO | RNF |
| $2\text{-}\mathcal{M}_0$ | GAT ($\mathcal{G}$) | $0.46 \pm 0.00$ | $\mathbf{0.46 \pm 0.00}$ | $0.47 \pm 0.01$ | $0.80 \pm 0.00$ | $\mathbf{0.80 \pm 0.00}$ | $0.80 \pm 0.00$ |
| | GCN ($\mathcal{G}$) | $0.46 \pm 0.00$ | $\mathbf{0.46 \pm 0.00}$ | $0.47 \pm 0.01$ | $0.80 \pm 0.00$ | $\mathbf{0.80 \pm 0.00}$ | $0.80 \pm 0.00$ |
| | MLP ($\mathcal{G}$) | $0.46 \pm 0.00$ | $\mathbf{0.46 \pm 0.00}$ | $0.46 \pm 0.01$ | $0.80 \pm 0.00$ | $\mathbf{0.80 \pm 0.00}$ | $0.80 \pm 0.00$ |
| | TAG ($\mathcal{G}$) | $0.46 \pm 0.00$ | $\mathbf{0.46 \pm 0.00}$ | $0.46 \pm 0.01$ | $0.80 \pm 0.00$ | $\mathbf{0.80 \pm 0.00}$ | $0.80 \pm 0.00$ |
| | UniMP ($\mathcal{G}$) | $0.46 \pm 0.00$ | $\mathbf{0.46 \pm 0.00}$ | $0.46 \pm 0.01$ | $0.80 \pm 0.00$ | $\mathbf{0.80 \pm 0.00}$ | $0.80 \pm 0.00$ |
| | CellMP ($\mathcal{T}$) | $0.85 \pm 0.11$ | | $\mathbf{0.89 \pm 0.01}$ | $0.80 \pm 0.34$ | | $\mathbf{0.94 \pm 0.01}$ |
| | CT ($\mathcal{T}$) | $\mathbf{0.91 \pm 0.01}$ | | $0.69 \pm 0.15$ | $\mathbf{0.94 \pm 0.00}$ | | $0.92 \pm 0.01$ |
| | DECT ($\mathcal{T}$) | $0.45 \pm 0.00$ | $0.45 \pm 0.00$ | $0.45 \pm 0.00$ | $0.80 \pm 0.00$ | $\mathbf{0.80 \pm 0.00}$ | $0.80 \pm 0.00$ |
| | SAN ($\mathcal{T}$) | $0.54 \pm 0.10$ | | $0.67 \pm 0.16$ | $0.35 \pm 0.36$ | | $0.72 \pm 0.26$ |
| | SCCN ($\mathcal{T}$) | $0.85 \pm 0.08$ | | $0.66 \pm 0.03$ | $0.78 \pm 0.35$ | | $0.77 \pm 0.30$ |
| | SCCNN ($\mathcal{T}$) | $0.54 \pm 0.10$ | | $0.61 \pm 0.02$ | $0.11 \pm 0.00$ | | $0.26 \pm 0.07$ |
| | SCN ($\mathcal{T}$) | $0.37 \pm 0.12$ | | $0.50 \pm 0.04$ | $0.50 \pm 0.41$ | | $0.73 \pm 0.09$ |
| $2\text{-}\mathcal{M}_H^0$ | GAT ($\mathcal{G}$) | $0.48 \pm 0.00$ | $0.49 \pm 0.00$ | $0.48 \pm 0.00$ | $0.54 \pm 0.00$ | $\mathbf{0.54 \pm 0.00}$ | $0.54 \pm 0.00$ |
| | GCN ($\mathcal{G}$) | $0.49 \pm 0.00$ | $0.48 \pm 0.01$ | $0.50 \pm 0.02$ | $0.54 \pm 0.00$ | $\mathbf{0.54 \pm 0.00}$ | $0.54 \pm 0.00$ |
| | MLP ($\mathcal{G}$) | $0.49 \pm 0.00$ | $0.49 \pm 0.00$ | $0.48 \pm 0.01$ | $0.54 \pm 0.00$ | $\mathbf{0.54 \pm 0.00}$ | $0.54 \pm 0.00$ |
| | TAG ($\mathcal{G}$) | $0.49 \pm 0.00$ | $0.49 \pm 0.00$ | $0.49 \pm 0.01$ | $0.54 \pm 0.00$ | $\mathbf{0.54 \pm 0.00}$ | $0.54 \pm 0.00$ |
| | UniMP ($\mathcal{G}$) | $0.49 \pm 0.00$ | $0.49 \pm 0.00$ | $0.49 \pm 0.01$ | $0.54 \pm 0.00$ | $\mathbf{0.54 \pm 0.00}$ | $0.54 \pm 0.00$ |
| | CellMP ($\mathcal{T}$) | $0.63 \pm 0.14$ | | $\mathbf{0.82 \pm 0.00}$ | $0.19 \pm 0.03$ | | $\mathbf{0.73 \pm 0.00}$ |
| | CT ($\mathcal{T}$) | $\mathbf{0.83 \pm 0.01}$ | | $0.50 \pm 0.02$ | $\mathbf{0.74 \pm 0.00}$ | | $0.54 \pm 0.00$ |
| | DECT ($\mathcal{T}$) | $0.50 \pm 0.00$ | $\mathbf{0.50 \pm 0.00}$ | $0.48 \pm 0.01$ | $0.54 \pm 0.00$ | $\mathbf{0.54 \pm 0.00}$ | $0.54 \pm 0.00$ |
| | SAN ($\mathcal{T}$) | $0.49 \pm 0.10$ | | $0.59 \pm 0.10$ | $0.54 \pm 0.03$ | | $0.61 \pm 0.03$ |
| | SCCN ($\mathcal{T}$) | $0.80 \pm 0.00$ | | $0.65 \pm 0.05$ | $0.73 \pm 0.00$ | | $0.57 \pm 0.03$ |
| | SCCNN ($\mathcal{T}$) | $0.59 \pm 0.10$ | | $0.52 \pm 0.02$ | $0.51 \pm 0.12$ | | $0.55 \pm 0.01$ |
| | SCN ($\mathcal{T}$) | $0.53 \pm 0.11$ | | $0.49 \pm 0.06$ | $0.30 \pm 0.14$ | | $0.43 \pm 0.03$ |
| $2\text{-}\mathcal{M}_H^1$ | GAT ($\mathcal{G}$) | $0.41 \pm 0.03$ | $\mathbf{0.53 \pm 0.02}$ | $0.50 \pm 0.01$ | $\mathbf{0.54 \pm 0.00}$ | $\mathbf{0.54 \pm 0.00}$ | $0.54 \pm 0.00$ |
| | GCN ($\mathcal{G}$) | $0.42 \pm 0.04$ | $0.51 \pm 0.04$ | $0.50 \pm 0.01$ | $\mathbf{0.54 \pm 0.00}$ | $\mathbf{0.54 \pm 0.00}$ | $0.54 \pm 0.00$ |
| | MLP ($\mathcal{G}$) | $0.43 \pm 0.04$ | $0.49 \pm 0.06$ | $0.50 \pm 0.01$ | $\mathbf{0.54 \pm 0.00}$ | $\mathbf{0.54 \pm 0.00}$ | $0.54 \pm 0.00$ |
| | TAG ($\mathcal{G}$) | $0.42 \pm 0.04$ | $0.50 \pm 0.03$ | $0.43 \pm 0.01$ | $0.51 \pm 0.09$ | $0.33 \pm 0.15$ | $0.54 \pm 0.00$ |
| | UniMP ($\mathcal{G}$) | $0.45 \pm 0.03$ | $0.42 \pm 0.03$ | $0.41 \pm 0.01$ | $0.45 \pm 0.13$ | $\mathbf{0.54 \pm 0.00}$ | $0.54 \pm 0.00$ |
| | CellMP ($\mathcal{T}$) | $0.57 \pm 0.06$ | | $\mathbf{0.62 \pm 0.02}$ | $0.47 \pm 0.17$ | | $\mathbf{0.55 \pm 0.01}$ |
| | CT ($\mathcal{T}$) | $\mathbf{0.72 \pm 0.13}$ | | $0.49 \pm 0.01$ | $0.51 \pm 0.20$ | | $0.54 \pm 0.00$ |
| | SAN ($\mathcal{T}$) | $0.49 \pm 0.02$ | | $0.53 \pm 0.04$ | $0.48 \pm 0.18$ | | $0.54 \pm 0.00$ |
| | SCCN ($\mathcal{T}$) | $0.67 \pm 0.04$ | | $0.53 \pm 0.04$ | $\mathbf{0.54 \pm 0.00}$ | | $0.54 \pm 0.00$ |
| | SCCNN ($\mathcal{T}$) | $0.51 \pm 0.01$ | | $0.51 \pm 0.01$ | $\mathbf{0.54 \pm 0.00}$ | | $0.54 \pm 0.00$ |
| | SCN ($\mathcal{T}$) | $0.51 \pm 0.07$ | | $0.49 \pm 0.04$ | $0.35 \pm 0.18$ | | $0.53 \pm 0.03$ |

