# OpenReview forum: "MANTRA: The Manifold Triangulations Assemblage"
_ICLR.cc/2025/Conference — ICLR 2025 Poster_

### Official Review · Reviewer_ChKW · 2024-10-28

**Soundness:** 4
**Presentation:** 4
**Contribution:** 3
**Rating:** 8
**Confidence:** 3

**Summary:**

This work proposes a synthetic benchmark dataset for topological deep learning. Contrary to graphs representing pairwise interactions, this dataset focuses on higher-order interactions. It is generated by considering the simplicial complexes of 2 and 3-dimensional manifolds, resulting in structures with up to 10 vertices. Samples are labeled with topological descriptors, such as Betti numbers, torsion, orientability, genus, or homeomorphisms.

After introducing the dataset, several experiments are carried out to compare the performance of five graph and four topological learning models on topological prediction tasks. The main findings are that topological models perform better than graph models on these tasks, however, topological models are far from having a strong performance on these tasks.

**Strengths:**

- The paper is very well presented in terms of structure, rigor, and backing statements with relevant references or evidence.
- As far as I can tell, good conventions are followed to make the dataset easily available and usable.
- The experimental protocol is overall solid and the drawn conclusions are relevant and adequate.
- Limitations are discussed.

**Weaknesses:**

I have two minor remarks:
- The scope of the dataset is limited in dimension (order of interactions) and number of vertices, as well as being purely synthetic. While these are not bad aspects in their own right and this certainly is justified given that TDL is in its early days as it allows to study basic properties of models (e.g., remeshing invariance), the benchmark could potentially quickly lose its relevance.
- As the authors acknowledge, several non-message passing TDL architectures were not included in the experiments. As such, one of the most intriguing interpretations of the results is somewhat diminished ("suggesting that new approaches are needed to leverage higher-order structure associated with the topological information in the dataset").

**Questions:**

- What is your view on the scope of the dataset in terms of the vertices and dimensionality? I would suggest adding a short discussion on this in the limitations.
- I am not sure if I agree with the statement in line 204: "To ensure fairness, all configurations use the same learning rate of 0.01". Different models may work best with different learning rates, which likely requires cumbersome hyperparameter tuning. The authors acknowledge this as a limitation, but I would reconsider the phrasing of this sentence -- is this really a "fair" comparison?

---

> ### Author Response · Authors · 2024-11-20
>
> Dear reviewer, thank you for your positive review. Below we answer your weaknesses and questions.
>
> #### Weaknesses
>
> 1.  We view the dataset as an entry point for models that aim to leverage high-order features in data. We would be glad if the
>     dataset quickly loses its relevance, as this would indicate that topological models are becoming robust in predicting topology.
>     However, if this occurs, we can expand the dataset to include triangulations with up to 11 vertices, as specified in our response
>     to reviewer fkvZ. This would make the dataset significantly larger. In fact, we can make the dataset arbitrarily large through
>     subdivisions of triangulations, posing new challenges in computational efficiency for topological models.
>
> 2.  We are rerunning the experiments with some more models to address this concern. We expect to publish the first preeliminar results on surfaces by tomorrow on three more models, as specified in other reviewers' answers.
>
> #### Questions
>
> 1.  Our dataset is a complete enumeration of triangulations of closed, connected two- and three-dimensional manifolds up to 10 vertices. This number of vertices is similar to the average number of vertices in some datasets like molecular datasets (see OGB molecular datasets    \[1\] or ZINC \[2\]), which usually contain between 20 and 30 vertices, but is significantly smaller than the number of vertices in some other datasets, such as fine-grained mesh datasets. It might seem that transferring observations from experiments executed in MANTRA to other datasets would be challenging due to the difference in size between our dataset and some other ones. However, subdivisions can be applied to obtain simplicial complexes with an arbitrarily large number of vertices, allowing the graph algorithms to work on more similar domains.
>
>
>     Moreover, there is still room to add more triangulations. Specifically, using the same source we used for the original version of
>     MANTRA, we could extract some of the triangulations up to $n=14$, although we would face significant storage and enumeration
>     challenges since the number of triangulations grows rapidly with $n$ and, to the best of our knowledge, there are no complete
>     enumerations of triangulations for $n\geq 13$ available. See answer to Reviewer fkvZ for more details.
>
>
>     Regarding the dimension, we believe that our dataset compiles triangulations for the most relevant higher dimensions, namely 2 and
>     3, as most real-world data are one-, two-, or three-dimensional (e.g., molecular graphs are 1D, meshes are 2D, cad reps can be up to
>     3D...). Furthermore, to the best of our knowledge, enumerations of triangulations in dimensions greater than 3 (and even in
>     dimension 3) are still active areas of research.
>
>
>     In general, our code implementations and data are flexible enough to develop additional variants of the dataset: the large-scale
>     version proposed to Reviewer fkvZ, or a dataset containing only minimal triangulations, that is, triangulations that are not subdivisions
>     of other triangulations. Thus, we firmly believe that our dataset, although limited on the number of vertices and dimensions, is a good
>     starting benchmark dataset to test high-order models, covering topological invariants of the most-abundant type of high-order
>     structured data, namely surfaces and volumes. We have included a paragraph on the limitations, summarizing our response.
>
> 2.  You are right and we will rephrase the sentence. To be completely fair, we should perform a grid search of several hyperparameters, what we did not do due to lack of resources and lack of computational efficiency of existing implementations.
>
> \[1\] Hu, Weihua, et al. \"Open graph benchmark: Datasets for machine learning on graphs.\" Advances in neural information processing systems 33 (2020): 22118-22133.
>
> \[2\] Sterling and Irwin, J. Chem. Inf. Model, 2015 http://pubs.acs.org/doi/abs/10.1021/acs.jcim.5b00559

---

> ### Comment · Reviewer_ChKW · 2024-11-25
>
> I thank the authors for their detailed response. I do not have further questions or remarks, apart from a minor comment:
>
> While I understand this might be important for testing TDL in very controlled environments, the complete enumeration of triangulations has less significance from the perspective of many applications. For geometric meshes, for example, handling much larger triangulations is of much greater importance instead. While I understand this can be mimicked with subdivisions, this is not the primary focus of the submitted work. While I appreciate the rigor, overfocusing on this synthetic aspect poses the danger of creating a gap to relevant applications that the TDL methods should ultimately tackle.

---

> > ### Author Response · Authors · 2024-11-25
> >
> > You raised an important concern in your comment, and we agree with you: our dataset does not cover the full spectra of data properties present in some datasets, such as the variation on size and coordinates of geometric meshes, that a robust of TDL should handle properly. For this reason, we are updating the limitations section to state this precisely. Thank you very much for engaging with us again. Still, we want to emphasize that our dataset is not synthetic, yet small in the number of vertices, in the sense that we are taking all the possible simplicial representations of manifolds that appear in  real-life (surfaces, solids, etc.) applications.

---

### Official Review · Reviewer_1p1z · 2024-10-30

**Soundness:** 2
**Presentation:** 2
**Contribution:** 1
**Rating:** 3
**Confidence:** 4

**Summary:**

The paper aims to benchmark the performance of graph-based and simplicial complex-based models on a newly-introduced 2-/3- manifold triangulation dataset. The paper focused on topological feature prediction tasks, including predicting Betti numbers, homeomorphism type and orientability.
Overall, the research question is interesting but the paper requires more work to be considered for publication. Please see my detailed comments and questions below.

**Strengths:**

The research question is interesting. It is novel to investigate the performance of graph and high-order network models on predicting topological features.

**Weaknesses:**

- There are various works on estimating Betti numbers, but the authors did not mention them. They should provide as solid baselines. For example:
  - Estimating Betti Numbers Using Deep Learning
  - A (simple) classical algorithm for estimating Betti numbers
- The experiments are rather unclear, for example, details on how the architectures are not provided. As a paper in datasets and benchmarks, this is necessary and as a reviewer, I'm interested in knowing how the architectures are designed and if they are reasonable.
  - Some experimental setup choices are not well-justified. For example, why randomly generated scalars are used as input features for graph-based models and randomly generated eight-dim vectors are used as feature vectors for simplicial complex-based models?
- Authors made a few claims which are not well-supported by either the experiments or literature.

Note: Please see my detailed questions below related to these points.

**Questions:**

1. about claims in the paper: 1) (058-060) To the best of our knowledge, the only publicly-available high-order dataset is the “Torus” dataset proposed in Eitan et al.(2024), which consists of a small number of unions of tori triangulations. 2) (069-071) The authors claimed this is the __first__ instance of a large, diverse, and intrinsically high-order dataset, comprising triangulations of combinatorial 2- and 3-manifolds.

  - Authors refer to datasets that contain nontrivial higher-order structures as higher-order datasets. It's worthwhile to point out that such structure and the datasets have been long- and well-studied by network science community, as well as numerical modeling, and there has been numerous efforts in building and modeling higher-order topological structures in various networks. However, there are way more such datasets as far as I'm aware of.
    - In computation biology and social networks, such datasets with higher-order structures appear very often.
    - in numerical modeling, computational geometry, 2D and 3D manifold triangulations are almost everywhere and they easily grow to the order of millions.
    - In topological deep learning community, such datasets have been applied by different papers as well, e.g., 1) SaNN: Simple Yet Powerful Simplicial-aware Neural Networks; 2) Unsupervised Parameter-free Simplicial Representation Learning with Scattering Transforms.

  - As a reviewer, I really shouldn't provide a more complete list of such literatures. I believe this paper could benefit from a more thorough literature review. Note that _there is no such dataset_ does not work as a strong motivation. Moreover, this work is hardly the __first__  dataset involved with high-order topological structures.

2. about the experiment tasks: as a researcher in TDA, I would like to see how graph- and simplicial complex-based models compare to the TDA methods for the investigated three tasks in this paper. Again, this is necessarily needed for being a datasets and benchmarks paper.

3. I do not agree with authors *define* the lower/upper degree of a simplex as the number of simplices which share a face/coface with it. This number is merely the number of neighboring simplices, but not the degree for simplices. The _degree_ of a simplex is the number of faces/cofaces it has. Please refer to _Spectra of combinatorial Laplace operators on simplicial  complexes_ and _RANDOM WALKS ON SIMPLICIAL COMPLEXES AND THE NORMALIZED HODGE 1-LAPLACIAN_ for such definitions.

4. Confusing experimental setup:
   1. Randomly generated scalars are used as feature vectors for graph-based models and randomly generated eight-dim vectors are used as feature vectors for simplicial complex-based models?
      - Questions: What is the motivation of using randomly generated values as input features? Any useful information do they provide to the models or tasks?

   2. In predicting 0-betti numbers, you only have label 1. When seeing results on predicting $\beta_0$, I have the following questions:
      -  If graph-based models have 100% accuracy, how come simplicial-complex based models can be poor not nearly 50%? My question comes from the fact that: simplicial complex based models are __generalizations__ of graph-based models. If GAT/GCN has an accuracy of 100%, then SAN/SCCN, which are really extensions to higher-order simplices, should be able to achieve the same accuracy _when using the same output_ (Note that here I assume authors considered using the same type of output for predictions, e.g., when using node representations of GCN for predicting $\beta_0$, one should also use node representation of SCCN for predicting $\beta_0$).
      -  On the other hand, most of the models implemented in this paper are performing representation learning, there is a need for a classifier or a regressor to predict the labels. Here it matters how a classifier/regressor, which takes the learned representation and computes the needed output, is defined. Yet, I couldn't find any details on this, but only __readout__ layers of different models, which are either mean, or sum of all embeddings. I wonder if authors simply used the readout to predict topological features. It doesn't make much sense to apply a direct readout to predict different topological features.
   3. what is the dataset $2-\mathcal{M}_H^1$? I tried to check the appendix but couldn't find it. If I guessed correctly, it is a refined version of $2-\mathcal{M}_H^0$. But how is this refinement performed?
   4. I appreciate the authors'effort in investigating the topological invariance of the models. This however is not well-analyzed in the paper, yet, the authors made the claim that high-order MP-based models are not invarinat relative to topological transformations ... in line 284.
      1. First of all, how is topological transformation defined? Is it only refinement of the triangulation?
      2. A proper topological predictor should be considered, yet this is unclear from the paper.
      3. I do not really see message-passing models are implemented. By that, I mean Bodnar et al., 2021a.


other minor suggestions:
- line 150: It seems incorrect to say something was degraded. I'm not an English major, but I think it should be "someone was degraded" or "something degraded". Please ignore it if I'm wrong.
- Please figure out a way to present Tables 9-22 without rotating them 90 degrees. Though I tried reading them, they are still not easy for readers.

---

> ### Author Response · Authors · 2024-11-20
>
> Dear reviewer, thank you very much for your thorough review. Let us
> address the weaknesses you pointed out below.
>
> 1.  We are grateful for the references you provided here. We are going
>     to add them to our revised version of the paper, as they are indeed
>     important for the problem at hand. However, we want to mention that
>     the approaches of both papers are fundamentally different because:
>
>     1.  in the first case, deep learning is applied to point cloud
>         samples of manifolds, which forget the topological structure of
>         the manifolds, in contrast to our dataset where the full
>         topological structure of the input data is preserved by the
>         triangulation;
>
>     2.  in the second case, a classical approximation algorithm is used
>         to approximate Betti numbers, that is fundamentally different to
>         study if neural networks can predict these quantities robustly.
>         In fact, our results suggest that, as for now, classical
>         approximation algorithms are better-suited to analyze the
>         topology of simplicial complexes, at least when it comes to
>         standard tasks such as Betti number calculations.
>
> 2.  Due to space constraints, we added hyperparameter details of the
>     architectures in Appendix B, and we cited the relevant architecture
>     papers in the main text. However, we have expanded Appendix B to add
>     more details about the different architectures. Regarding the
>     suitability of the architectures used, graph architectures were all
>     published in high-impact conferences and have been extensively
>     tested in the literature and two of three simplicial models were
>     already published in important venues, i.e. PMLR and IEEE PAMI and
>     also used in a TDL benchmark on graph datasets before \[1\].
>     Regarding the length of the random node features, we modeled it
>     after the maximum degree of the vertices, which for the $2$
>     manifolds is $8$. One hot encoding of the degree yields an
>     $8$-dimensional vector, and thus we took the same length for random
>     features. This is an arbitrary decision, because the idea of random
>     features is to let the model learn that features carry no meaning on
>     the input data.
>
> 3.  We tried to moderate our tone when stating our conclusions and
>     claims, and to justify our conclusions in the best way we could
>     think of when writing the paper. Regarding graph vs high-order
>     models, we observed what was accepted as a fact in the community:
>     simplicial models were better than graph models for simplicial
>     datasets. This was further justified in the paper by the theoretical
>     fact that different surfaces can have the same graphs and thus graph
>     neural networks are unable to tell the difference between the
>     triangulations because you need the triangle information to discern
>     between both manifolds. Regarding overall performance of simplicial
>     models, we found that they (or rather their current implementations)
>     are not yet robust to remeshing invariance nor topological property
>     prediction, which was unexpected due to that, to the best of our
>     knowledge, the position paper \[5\] claimed that simplicial models
>     were remeshing invariant. We tried to check this claim, but,
>     surprisingly, it did not hold. Still, we are completely open to
>     discuss specific claims and to rewrite them according to the
>     discussion. The main objective of this paper is not to categorically
>     claim properties of graph or simplicial models, but to provide a new
>     dataset that allows researchers to go deeper on the questions we
>     posed in our paper, such as the remeshing invariance property. This
>     means that we are completely open to adjust our claims after a
>     proper discussion of them.

---

> > ### Author Response · Authors · 2024-11-20
> >
> > Now let us address the questions you pointed out below:
> >
> > 1.  Regarding the datasets, we are aware of attempts to convert
> >     (lift) originally graph datasets into high-order datasets by means
> >     of adding simplices/cells to relationships between more than two
> >     vertices. However, in most cases, this extra information can be
> >     completely inferred from the graph structure, making cells redundant
> >     in the data. To the best of our knowledge, there has not been any
> >     attempt to prove that these extra relationships cannot be
> >     inferred solely by means of GNNs. For example, in your first cited paper (1) two of three
> >     datasets are graph based, and the third one (simplicial-closure
> >     prediction), although include simplices, the simplices need, by
> >     nature, to be annotated (e.g., in the case of coauthorship
> >     data, by definition, if you do not annotate a simplex, then you
> >     cannot have two different papers coauthored by the same authors
> >     because there can be only a simplex per set of authors). Given the
> >     annotations, though, you can simply annotate the edges associated
> >     with the simplex in the graph part of the network, and thus,
> >     retrieving the simplex information is as easy as looking for cliques
> >     and matching annotations. In our case, you cannot distinguish the
> >     simplicial complexes using solely the information from their graphs,
> >     making mandatory for the performant networks to use the high order
> >     information of the triangles. We attached a theoretical
> >     justification in the paper. In the case of the cited paper (2),
> >     three datasets are (hyper)graph-based and the other one comes from the same source as the simplicial
> >     dataset in (1). Still, we strongly agree that the paper \[2\] is
> >     relevant, and we have added this paper to our literature
> >     review in the revised version. Regarding
> >     our justification about the need of high-order datasets and their
> >     scarcity, two examples of the importance of these datasets can be
> >     seen in (1) \[1\], where they simply lift graph datasets, and (2) in
> >     the position paper \[3\], where it is argued that developing more
> >     high-order datasets is fundamental for the field of topological deep
> >     learning (Open problem 2).
> >
> > 2.  We did not include TDA methods because persistent homology can
> >     obtain deterministically the labels used
> >     in our experiments using a large enough coefficient field. This is
> >     because with a constant filtration, persistent homology recovers the
> >     homology of a simplicial complex, and thus we obtain orientability,
> >     Betti numbers, and the homeomorphism type for triangulated surfaces.
> >     Still, we are executing the same experiments with the topological model defined in \[6\] for completeness.
> >
> > 3.  You are right. We have renamed our upper/lower degree in the revised version.
> >
> > 4.  Subquestions answered below:
> >
> >     1.  By introducing random features, we discourage the network from
> >         relying on feature vectors and instead encourage it to focus
> >         solely on the structure of the triangulations, which is the only
> >         relevant information for this problem. Unfortunately, models
> >         often require feature vectors by design. Using random features also allows
> >         us to avoid providing extra manually computed information that
> >         could enable the network to infer the correct labels without
> >         utilizing the input's high-order structure.
> >
> >     2.  We see one main justification here: although you can set the
> >         weights appropriately to avoid the features from the high-order
> >         simplices, in practical scenarios both types of networks do not
> >         follow the same optimization path, meaning that the functions
> >         represented at the end of the training are different. Still, it
> >         is a remarkable finding in our experiments, and
> >         further experiments should be performed to completely understand
> >         this phenomenon. As this is mainly a dataset paper and a first
> >         preliminary study on it, we think that further experiments are
> >         out of scope for this manuscript and should be performed in a
> >         follow-up paper.
> >
> >     3.  Readout is the standard way to extract a graph/simplicial
> >         prediction from features on nodes/simplices, and that is why we
> >         adopted this approach. More involved ways, as in the previous
> >         justification, belong to a follow-up paper doing a
> >         deeper study on our findings.
> >
> >     4.  The refinement is performed using barycentric subdivision of
> >         abstract simplicial complexes. For the revised version of the
> >         paper, we will explicitly state this definition.
> >
> > 5.  The topological transformation is a barycentric subdivision
> >     of the simplicial complex. The beginning of the definition
> >     can be found in line 578 of the current manuscript. The three
> >     simplicial complex models are message passing architectures as
> >     described in \[4\] and in two papers of the three architectures.

---

> ### Author Response · Authors · 2024-11-20
>
> Regarding the minor suggestions, we will try to change in some way the
> presentation of the tables, as we agree that they are hard to read being
> rotated. Regarding the use of degraded, we have rephrased the whole
> sentence avoiding its use.
>
> These are the references used in our answer(s).
>
> \[1\] Telyatnikov, Lev, et al. \"TopoBenchmarkX: A Framework for
> Benchmarking Topological Deep Learning.\" arXiv preprint
> arXiv:2406.06642 (2024).
>
> \[2\] Benson, Austin R., et al. \"Simplicial closure and higher-order
> link prediction.\" Proceedings of the National Academy of Sciences
> 115.48 (2018): E11221-E11230.
>
> \[3\] Papamarkou, Theodore, et al. \"Position: Topological Deep Learning
> is the New Frontier for Relational Learning.\" Forty-first International
> Conference on Machine Learning. 2024.
>
> \[4\] Papillon, Mathilde, et al. \"Architectures of Topological Deep
> Learning: A Survey of Message-Passing Topological Neural Networks.\"
> arXiv preprint arXiv:2304.10031 (2023).
>
> \[5\] Papamarkou, Theodore, et al. \"Position: Topological Deep Learning
> is the New Frontier for Relational Learning.\" Forty-first International
> Conference on Machine Learning. 2024.
>
> \[6\] Röell, Ernst, and Bastian Rieck. \"Differentiable Euler
> Characteristic Transforms for Shape Classification.\" The Twelfth
> International Conference on Learning Representations.

---

> > ### Comment · Reviewer_1p1z · 2024-11-22
> >
> > Thank you for the response and the additional experiments. However, I respectfully disagree with the authors on the following points:
> > ##### dataset:
> > The goal of this paper is to introduce higher-order topology dataset.
> > While I provided two examples of already available dataset, I meant to say there are way more datasets available in the literature. It's clear that "torus" is NOT the only available purely high-order dataset. In biology related works, there are many datasets which are involved with higher-order structures.
> > - Clustering-independent analysis of genomic  data using spectral simplicial theory.
> > - The topology of higher-order complexes associated with brain hubs in human connectomes.
> > - Two’s company, three (or more) is a simplex.
> >
> > Similarly, in other fields, such as social networks,
> > - A novel simplicial complex representation of social media texts: The case of Twitter.
> >
> > other machine learning paper used higher-order data (high-dim convex hulls):
> > - Clifford Group Equivariant Simplicial Message Passing Networks.
> >
> > Many datasets are not lifted from graphs, but are directly built on higher-order structures. Again, this literature review should be carried out beforehand by authors. I dont see the contribution of this newly introduced dataset.
> >
> >
> > ##### random features as inputs:
> > 1. Using random vectors as input does not make sense. When performing any kinds of learning, you have inputs which are informative to the goal/output either directly or indirectly. In your case, if one has a triangulation of a ball and aims to predict the Betti number, so using inputs as totally random numbers, you are able to train a model to predict this number?
> >    - How can this randomly-trained model be useful for downstream tasks, if you dont do unlimited training? If input $[1,2,3]$ is trained well, is your model able to predict correctly using input $[1,1,1]$? Please think about how expressive the model needs to be such that it takes any random input and still is able to predict the Betti number.
> >    - You have much available information which can be used as inputs, e.g., the coordinates of the points, the degrees, etc.
> >
> > ##### Readout layers:
> > I understand you took the standard way for readout. But readout is a summary of the representations of the object learned by the model. You have a follow-up task to predict or classify, and you would need a predictor or classifier which takes this summary (readout) as inputs and obtains the output for either predictions or classifications. For example, for link prediction, you need a classifier to classify whether two nodes are connected or not, and this classifier takes the readout of some GNN models as input. My question is what kind of classifier you used for predicting the topological features. If you take the mean/sum of the final embeddings, how do you obtain the betti numbers from this?

---

> > > ### Author Response · Authors · 2024-11-23
> > >
> > > Dear reviewer,
> > >
> > > We think that there is a misunderstanding of what our dataset entails and respectfully disagree with your stance that this paper does not contribute anything.
> > >
> > > More specifically, we find that the other papers you mention cover different, distinct, aspects:
> > >
> > > In the first paper, it is specifically written that "in this paper, we address the problem of unsupervised feature selection in point clouds". The simplicial complexes they build are lifted from the point clouds, using Cech, Vietoris-Rips complexes, or similar. These complexes are not guaranteed to respect the topology of the manifold and the high-order information they contain depend on the construction they do.
> > >
> > > In the second paper, as well as in the source of the data they use ("Functional geometry of human connectomes"), they explicitly mention that they work with clique complexes, that are simplicial complexes built from graphs by adding all the cliques (and thus, simplices can be recovered from the graph algorithmically).
> > >
> > > In the third paper, the authors use clique complexes and independence complexes, that can be captured with graphs/hypergraphs. In the case of concurrence complexes, it is not clear that a bipartite graph cannot represent the same information as the Dowker complex.
> > >
> > > In the fourth paper, a lift is applied from a point cloud of topics.
> > >
> > > In the fifth, paper,  the CMU Human Motion Capture is a graph dataset, MD17 is a molecular (graph) dataset, and the dataset from the "NBA players 2D trajectory prediction" is also a graph dataset.
> > >
> > > With this, we are not saying that using topological structures in these datasets is not useful, as they add pertinent explicit bias on the relations of the data. However, **these relations are forged by hand, and are not naturally occurring in the data, so we argue that our dataset can be better suited to test if models capture and utilize high-order data that cannot be recovered directly from the graph structure either, deterministically or performing inference**. Also, even assuming that our dataset is not novel, which we consider it is because it provides the first dataset of purely combinatorial high-order simplicial complexes, we do not think that a dataset is not interesting because there are other similar datasets. If so, there would be only one dataset of images in the computer vision community, but it turns out that there are many of them, each one with its unique characteristics, as it happens in our case. Using the same argument, all the datasets explored in the papers you cited are redundant to check the performance of a deep neural network in a high order dataset, still the papers use more than one to check the capacity of their algorithms. We believe there to be value in diversity of datasets.
> > >
> > > **Random Node Features**
> > >
> > > When it comes to the random node features, you are absolutely right to observe that other datasets typically come with informative features. When developing and evaluating machine learning methods  on such datasets it therefore becomes difficult to asses the efficacy of the "higher order" part of the method. For instance, when the node features are informative,  it could very well be that viewing the dataset as a point cloud will be sufficient. Hence our dataset serves partially as an ablation tool for assessing the efficacy of methods that leverage higher order structures, such as graphs and simplicial complexes. When the features are random, i.e. non-informative, the higher order structures are the only informative part and if a model is able to learn in that case, it implies that the method actually leveraged the higher order structure in the dataset as compared to "just" using the nodes. The ability to do this ablation is one of the core contribution of our dataset.
> > >
> > > **Readout layers:**
> > >
> > > We emphasize that the representations learned by the model are shaped by the optimization objectives. We minimize the loss functions corresponding to the classification of Betti numbers, orientability, or homeomorphism type. Consequently, the learned representation of an input object is densely packed with information sufficient for these classification tasks. Given this design, no additional classifier is necessary, as a simple linear forward layer suffices to map the learned representations to the target objectives. For regression tasks, the model's final layer (feedforward) outputs feature vectors with dimensionality matching the number of regression targets (e.g., 3-dimensional vectors for 3 Betti numbers). For classification, we employ a similar strategy: in binary classification tasks, the final layer outputs one-dimensional features, and a standard decision boundary is applied. For multiclass classification, the final layer outputs feature vectors with as many dimensions as there are classes, and the predicted class is determined by the argmax of these outputs. The loss functions used are standard: cross-entropy for classification and MSE for regression.

---

> > > > ### Comment · Reviewer_1p1z · 2024-11-25
> > > >
> > > > 1. I respectully disagree.
> > > >    - All the complex datasets have the goal to capture the higher-order relations in the data beyond pairwise edges. All the papers I listed are involved with REAL higher-order structure, such as _multiple brain regions or neural units activated together_. __Note that these are not forged by hand. They exist in real applications. (Building higher-order network models is one of the main direction in the whole community of Network Science.)__ These work did build a complex model to model these actual structures. Despite they built clique complexes, one can easily extract simplicial complexes from them by merely removing some of them, based on, e.g., their weights or so, akin to how you build a kNN graph (conceptually extracted from the fully-connected complete graph, i.e., a 1-clique complex). Again, this is not a complete list of papers but to show there are tremendous works on extracting/modeling higher-order structures from real-world applications. However, they are completely missed from the work. In the paper _Clifford Group Equivariant Simplicial Message Passing Networks_, please have a look at the (synthetic) high-dim convex hulls (simplices). This is a comparable type of dataset to the ones you have. I want to kindly remind authors of that network science has been a much longer research topic than learning on graphs. When you work on datasets involved with higher-order structures, you should be aware of works beyond papers on learning on graphs.
> > > >    - __Evaluating various graph and higher-order learning models on simplicial complexes built from these datasets involved with REAL higher-order relations is much more meaningful__. I get your claim that this is the first combinatorial dataset, but it's more of the question if it brings values to the community. In the end, it boils down to the question if using this dataset can help to evaluate if one model using the higher-order structure is better than the one using the graph structure. __However, there are so many such real-world datasets as I listed above, which really involve higher-order structures__. Beyond those, as pointed out in my initial review, please have a look at also computer graphics for geometry processing and numerical modeling literature where triangulation of manifolds is ubiquitous.
> > > >
> > > >
> > > > 2. Your arguments on ```random features``` remain unjustified. If you truly want to check if only structures are enough for the predictions, why not using features that actually encode the structure? e.g., the coordinates; if you dont have coordinates, like the data-structure (using simplex tree) for representing simplices.; or certain fixed knowledge like degrees of simplices. Random inputs are really not helping to evaluate and hard to control. You wouldn't be able to obtain robust results or conclusions without infinite number training using different random inputs, especially when involved with high-dimensionality.
> > > >
> > > > 3. The optimization objective is of course for classification and supposed to be. That does not justify the methodology of constructing the prediction models using only the representation learning part. Many of the models are originally proposed for representation learning. It's a common practice to append a classifier/regressor to the learned representations to predict the target. For this reason, I remain skeptical about the methodology of the experiments.

---

> ### Author Response · Authors · 2024-11-26
>
> Dear reviewer,
>
>
> 1. We think there is a misunderstanding between us. We are not saying that there is no high-order information in complex networks or other applications aside algebraic topology. In fact, we believe that the whole point of topological deep learning is to be useful in scenarios like science, complex networks, topology, etc. What we are arguing is that, although the encoding of the high-order information as simplices/cells is useful, it is not clear that the representations of the current datasets, which contain labels, cannot be learned and handled by graph models without the need of high-order models. In your example, a non-high-order deterministic algorithm could retrieve the high-order information by simply looking at the cliques of the graph and then filtering them by weights. This is probably not the most efficient way of learning from the complex network data, but it is certain a way of retrieving information using only the graph information. This is not possible in our dataset, as several simplicial complexes share the graph part, and thus successful models in MANTRA need to use the high-order information of the simplicial complexes.  Regarding the complex hull dataset, the complex hull is inferred from a sample of points, so you do not need high-order information. In contrast, manifolds are naturally occurring, non-synthetic, structures in topology materialized in real-world phenomena (like the manifold hypothesis) with a lot of relevance in (modern) pure and applied (low-dimensional) topology; see for example, the series of papers on manifold triangulations cited [1-7].  With this, we want to stress out that triangulations are important and meaningful in its own field, although they are not part of the community of network science.
> Since we deeply care about the quality of our work and the scholarship aspects, we are adding all the dataset references in complex networks you cited in your answers in the introduction.
> 2. We actually used our version of degrees of simplices, that contains structural information about the simplicial complexes.
> 3. This is a standard approach in the graph and simplicial learning community. For instance, refer to the "Preliminaries" section of [8] (equation 2.4). More involved readout methods, such as those listed in [9], or more sophisticated pooling techniques beyond our mean pooling, as in [10], can lead to better results, but we think a further exploration of the dataset is out of scope and must be performed in follow-up papers.
>
> [1] Gunnar Brinkmann and Brendan D. McKay. Fast generation of planar graphs. MATCH Commun. Math. Comput. Chem. 58, 323-357 (2007).
>
> [2] Frank H. Lutz. Enumeration and random realization of triangulated surfaces. Discrete Differential Geometry (A. I. Bobenko, P. Schröder, J. M. Sullivan, and G. M. Ziegler, eds.). Oberwolfach Seminars 38, 235-253. Birkhäuser, Basel, 2008.
>
> [3] Thom Sulanke and Frank H. Lutz. Isomorphism-free lexicographic enumeration of triangulated surfaces and 3-manifolds. Eur. J. Comb. 30, 1965-1979 (2009).
>
> [4] Mark N. Ellingham and Chris Stephens. Triangular embeddings of complete graphs (neighborly maps) with 12 and 13 vertices. J. Combin. Des. 13, 336-344 (2005).
>
> [5] Frank H. Lutz. Triangulated Manifolds with Few Vertices: Geometric 3-Manifolds. Preprint, 48 pages, 2003; arXiv:math/0311116.
>
> [6] Frank H. Lutz. Triangulated surfaces and higher-dimensional manifolds. Oberwolfach Reports 3, No. 1, 706-707 (2006).
>
> [7] Frank H. Lutz. Periodic foams and simplicial manifolds with small valence. Oberwolfach Reports 4, No. 1, 228-230 (2007).
>
> [8] Xu, Keyulu, et al. "How powerful are graph neural networks?." arXiv preprint arXiv:1810.00826 (2018).
>
> [9] Navarin, Nicolò, Dinh Van Tran, and Alessandro Sperduti. "Universal readout for graph convolutional neural networks." 2019 international joint conference on neural networks (IJCNN). IEEE, 2019
>
> [10] Ying, Zhitao, et al. "Hierarchical graph representation learning with differentiable pooling." Advances in neural information processing systems 31 (2018)

---

> > ### Comment · Reviewer_1p1z · 2024-12-02
> >
> > Dear authors, I maintain my current rating. Unfortunately, my primary concern remains regarding the limited contribution of introducing a new high-order synthetic dataset. As mentioned multiple times, there are a great deal of datasets that involve __intrinsic__ higher-order network structures, arising from
> > 1. various real-world networks (in biology, chemistry, social networks, etc.) where different modeling methods and learning algorithms for higher-order structures have been one of the main research directions in network science and other communities within the specific application domains,
> > 2. triangulations of numerous real-world and synthetic geometries (particularly in computational geometry, e.g., by CGAL) where, for example, the computational graphics field strives for the efficient computation and processing methods on these large-dim triangulations.
> >
> > Thanks for clarifying the other two questions, though my concern on the use of random input features remains. I believe there should be a robust investigation on this, among other possible inputs.

---

> > > ### Author Response · Authors · 2024-12-02
> > >
> > > Thanks for your response. While we respect your decision, we want to point out that we believe that diversity in datasets is a crucial factor in improving models and our understanding of them.
> > >
> > > We therefore are confident that our dataset, which comes with provable correctness guarantees and whose construction as well as provenance are described in detail, provides added value to the community.

---

### Official Review · Reviewer_fkvZ · 2024-11-01

**Soundness:** 3
**Presentation:** 3
**Contribution:** 3
**Rating:** 6
**Confidence:** 4

**Summary:**

The paper introduces MANTRA, a new dataset designed for benchmarking topological deep learning (TDL) models. MANTRA consists of triangulations of two- and three-dimensional manifolds, labeled with various topological invariants such as Betti numbers, Homology torsion, homomorphism type (four options for 2D manifold and five options for 3D manifolds), and more. Using this dataset, the authors benchmark standard graph methods and TDL models on three types of tasks: Betti number prediction (2D + 3D), homeomorphism type prediction (2D), and orientability prediction (2D). Their results suggest that while TDL models are better at predicting higher order topological properties, they struggle with invariant to topology preserving transformations. The latter is demonstrated through additional experiments where barycentric subdivision -- a form of remeshing that preserves topological properties -- is performed on the test set.

**Strengths:**

- MANTRA is a large dataset specifically designed for topological property prediction, filling an important a gap in the TDL literature. Existing datasets are limited in scope, size, topological diversity and often rely on artificial topological lifting of graph data.
- The paper presents systematic analysis of graph methods vs. topological methods on fundamental topological property prediction tasks, establishing important baselines for future research in TDL.
- TDL models' ability to capture topological properties (**Q2** in the paper) is a fundamental question in TDL that remained under-explored to this point. Empirical results presented in the paper suggest that TDL models are better, compared to graph based methods, at learning topological properties from data.
- The question of invariance to topology preserving transformations (**Q3**) is also under-explored (albeit less clearly motivated). This paper demonstrate TDL models' sensitivity to remeshing by barycentric subdivisions -- a simple topology preserving transformation.
- The paper is clearly organized, with well defined questions, answered in a methodical manner.
- The authors ensure reproducibility by providing well-documented code repos, versioned datasets, and multiple data format options (raw and PyTorch Geometric), making it easy to build upon this work.

**Weaknesses:**

- **Topological diversity**
  - Other than the genus, first Betti number, and homeomorphism type for 2D manifolds, the distribution of topological invariants is limited and unbalanced.
    - For 3D manifolds:
       - Most Betti numbers, homeomorphism types, and torsion subgroups have only one or two possible values.
       - When two values do exist, one value represents over 99% of the cases.
  - All triangulations are limited to 10 or fewer vertices.

- **Limited model coverage**
  - The only TDL baselines included in the evaluation are simplicial complex models, excluding:
      - Cellular complex methods (e.g. [1, 2]).
      - Combinatorial complex methods (e.g. [3]).
      - Hypergraph methods (e.g. [4, 5]).
  - For both TDL and graph based methods, the evaluation only includes message-passing based architectures, excluding non-message passing graph neural networks (e.g. [6, 7, 8, 9]) and attention based TDL models (e.g. [3, 10]).
  - It would be informative to report the best performing model for each class in the main text.

- **Training methodology**
  - Models were trained for only 6 epochs, hyperparameters were not optimized. The can possibly lead to underestimation of the fully optimized performance of some of the architectures, limiting the scope of the conclusions that can be drawn from the experiments

---

[1] Bodnar et al. "Weisfeiler and Lehman Go Cellular: CW Networks", NeurIPS 2021.

[2] Hajij et al. "Cell complex neural networks", Topological Data Analysis and Beyond Workshop NeurIPS 2020.

[3] Hajij et al. "Topological Deep Learning: Going Beyond Graph Data", arXiv:2206.00606

[4] Feng et al. "Hypergraph neural networks", AAAI 2019.

[5] Kim et al. "Equivariant hypergraph neural networks." ECCV 2022.

[6] Maron et al. "Provably powerful graph networks." NeurIPS 2019.

[7] Abboud et al. "The surprising power of graph neural networks with random node initialization", arXiv:2010.01179.

[8] Bouritsas et al. "Improving graph neural network expressivity via subgraph isomorphism counting", IEEE PAMI 2022.

[9] Zhang et al. "A complete expressiveness hierarchy for subgraph gnns via subgraph weisfeiler-lehman tests", ICML 2023.

[10] Battiloro et al. "Generalized simplicial attention neural networks", arXiv:2309.02138.

**Questions:**

- **Dataset Generation**
  - How were the manifolds and triangulations generated?
  - What considerations did you make, if any, in order to encourage topological diversity in the dataset?

- **Experimental Results**
  - For the standard deviation results in Table 2, do they represent variation within a single model or across the entire class of models?

- **Computational Issues**
  - The authors cite computational constraints for limiting some aspect of the experiments, can the authors elaborate on the nature of these limitations given that all triangulations have 10 or fewer vertices?

---

> ### Author Response · Authors · 2024-11-20
>
> Dear reviewer, thank you for your review. Below, we answer to the
> weaknesses of the paper.
>
> #### Weaknesses
>
> 1.  Regarding topological diversity, we took all possible triangulations
>     up to ten vertices of surfaces and 3-manifolds. This means that the
>     diversity of labels on the surfaces and $3$-manifolds cannot be
>     improved unless we add more triangulations, as the label
>     distribution is the one occurring naturally in these topological
>     objects. Regarding the size of the dataset, we decided to stop at
>     $n=10$ because for $n=11$ there are $11,590,894$ and $172,638,650$
>     different triangulations for surfaces (from which we can gather
>     $219,321$ triangulations) and $3$-manifolds, rendering the dataset
>     too large (however, it is not impossible to retrieve triangulations
>     of $n=11$, see \[1\] and
>     <https://www3.math.tu-berlin.de/IfM/Nachrufe/Frank_Lutz/stellar/surfaces.html>).
>     We will be happy to add a large-scale version of the dataset if you
>     consider it necessary to the current paper before the camera-ready
>     revision.
>
> 2.  Initially, we chose only simplicial models from TopoModelX because
>     the cellular models can't leverage properties of cell complexes that
>     are not being leveraged by simplicial models, as there are no cells
>     that are not simplices in our dataset. Regarding equivariant models,
>     we avoided geometric equivariant models because MANTRA is purely
>     combinatorial, meaning that there are no geometric features on the
>     simplices such as coordinates. However, we have rerun the
>     experiments for the cellular models you referenced as \[1\] for a
>     more comprehensive benchmark. We expect to publish the first
>     preliminar results by tomorrow. Please, let us know if you consider
>     that there are more models that we should add to our benchmarks.
>
> 3.  We noticed that most graph-based models converged rather quickly
>     after only two or three epochs and that simplicial-based models
>     stopped converging after the same number of epochs. Hence, training
>     for more epochs is not likely to improve the results. Also, we
>     noticed a strong tendency for the models to predict constant
>     outputs, suggesting that hyperparameter tuning will most likely not
>     result in increased performance.
>
> #### Questions
>
> 1.  We extracted the triangulations from previous works of Frank H.
>     Lutz, that are gathered in the following webpage:
>     <https://www3.math.tu-berlin.de/IfM/Nachrufe/Frank_Lutz/stellar/>.
>     We have explicitly specified this in the revised version of the
>     manuscript, concretely, at the beggining of Section 2.
>
> 2.  They represent the variance across the entire set of models.
>
> 3.  Unfortunately, computational complexity does not originate from our
>     dataset, but from a combination of the simplicial model
>     implementations of the TopoModelX library, that are currently not
>     optimal, and the large sets of experiments performed (five random
>     seeds for each pair of dataset and model). Regarding model
>     efficiency, the library is under continuous development, and we
>     expect simplicial models to get faster with new versions of the
>     library. For a precise measurement of the slow-down, please refer to
>     Table 7 in our Appendix.
>
> \[1\] Sulanke, Thom, and Frank H. Lutz. \"Isomorphism-free lexicographic
> enumeration of triangulated surfaces and 3-manifolds.\" European Journal
> of Combinatorics 30.8 (2009): 1965-1979.
> \[2\] Hajij, Mustafa, et al. \"TopoX: a suite of Python packages for
> machine learning on topological domains.\" arXiv preprint
> arXiv:2402.02441 (2024).

---

### Official Review · Reviewer_r6xH · 2024-11-03

**Soundness:** 3
**Presentation:** 3
**Contribution:** 3
**Rating:** 6
**Confidence:** 3

**Summary:**

This paper introduces MANTRA, a large-scale, diverse, and high-order dataset of triangulations for benchmarking models in the topological deep learning space. The authors also assess graph and simplicial complex-based models on tasks like betti number estimation, homeomorphism type, and orientability detection. The introduction section is written well with enough definitions/context about various topological properties. The dataset specification includes all properties that are being computed for each object. Furthermore, nine different neural nets are evaluated on the classification task.

Overall, this paper could be a good contribution to topological deep learning, especially for developing advanced TDL methods and benchmarking them. However, the limiting factor is the evaluations (just MP networks) as they do not give a full picture of the impact of the dataset.

**Strengths:**

1. Extensive evaluation of currently available graph and simplicial complex-based models on MANTRA.

2. Provides a foundation for developing and benchmarking advanced TDL methods.

3. Insight into the ability of higher-order models to be invariant to triangulation transformations.

**Weaknesses:**

1. Evaluations are mostly focused on MP networks. It would help get better overall picture of the impact the dataset has by evaluating on other architectures like equivariant high-order neural nets and topological transformers.

2. The variance across results is surprising and I am not sure if running for more epochs/more hyperparameter tuning will address this.

**Questions:**

1. I'm not sure if you mention this in the paper but how are the initial labels and triangulations generated for each surface and three-dimensional manifold?

2. For a given surface, how much computational time is required for computing the fields mentioned Section 2?

3. Would MANTRA be useful for developing models for topological reconstruction (to recover incomplete surfaces or perform noise correction)? If so, how?

---

> ### Author Response · Authors · 2024-11-20
>
> Dear reviewer,
>
> Thank you for your review. We answer your two main concerns below:
>
> 1.  Regarding focusing on MP networks, this is mostly because current
>     simplicial models are of this nature. To the best of our knowledge,
>     there are only two papers on transformers for high-order domains:
>     one centered on cliques \[1\] not specifically tailored for
>     simplicial complexes (although they define how to work with them)
>     and another paper, still in review, studying general topological
>     transformers paper \[2\]. However, we are executing the experiments
>     with the model of \[2\] (that already incorporates concepts from
>     \[1\]). We expect to publish the first preliminary results by
>     tomorrow. Regarding equivariant neural networks, we decided not to
>     include them because MANTRA is purely combinatorial, meaning that
>     there are no natural geometric features. More specifically,
>     equivariant methods are useful if the simplicial complexes are
>     embedded in space, i.e. the vertices have coordinates. This is not
>     the case in MANTRA. However, if you have (a) specific network(s) in
>     mind, please, let us know and we will do our best to execute the
>     experiments before the discussion stage and camera-ready version.
>
> 2.  Regarding variance, there is hardly any variance in the results per
>     network and dataset. We acknowledge that current tables may lead to
>     confusion, as we use the variance across models and transforms in
>     them. We are working on a revision of our table presentation to make
>     this clear.
>
> Questions:
>
> 1.  (Q1 and Q2) The triangulations without barycentric subdivisions are
>     a complete enumeration of triangulations up to 10 vertices of closed
>     and connected two- and three-manifolds. Initially, these
>     triangulations and their labels were discovered, gathered and
>     published online by Frank H. Lutz in his websites:
>     <https://www3.math.tu-berlin.de/IfM/Nachrufe/Frank_Lutz/stellar/surfaces.html>
>     and
>     <https://www3.math.tu-berlin.de/IfM/Nachrufe/Frank_Lutz/stellar/3-manifolds.html>.
>     Labels can also be computed using deterministic algorithms from the
>     field of computational topology, specifically using algorithms to
>     compute homology groups that are $\mathcal{O}(m^3)$ with $m$ being
>     the number of simplices for their most basic implementation.
>     However, the computational complexity can greatly vary depending on
>     the algorithm and the boundary matrices, even achieving sublinear
>     complexity in some cases.
>
> 2.  We expect MANTRA to be useful as the first dataset on topological
>     reconstruction, as you can generate a new dataset by eliminating
>     certain simplices from either the original triangulations or from
>     the subdivision ones. Thanks to the code provided, it is easy to
>     generate this dataset. Also, due to the high quantity of
>     triangulations, we firmly believe that MANTRA will be a strong proof
>     of concept for papers tackling topological reconstruction.
>
> \[1\] Zhou, Cai, Rose Yu, and Yusu Wang. \"On the Theoretical Expressive
> Power and the Design Space of Higher-Order Graph Transformers.\"
> International Conference on Artificial Intelligence and Statistics.
> PMLR, 2024.
> \[2\] Ballester, Rubén, et al. \"Attending to Topological Spaces: The
> Cellular Transformer.\" arXiv preprint arXiv:2405.14094 (2024).

---

> > ### Comment · Reviewer_r6xH · 2024-11-24
> > **Response to Official Comment by Authors**
> >
> > I thank the authors for the rebuttal. I have increased my rating

---

### Author Response · Authors · 2024-11-21

Dear reviewers, find attached the results for the surface experiments without the barycentric subdivision. As soon as we have completed the experiments for the 3D datasets, we will update our manuscript with the complete and updated results. The models implemented are:

Cell MP (cellular message passing) [1]
Cellular Transformer (topological transformer) [2]
DECT (differentiable Euler characteristic transforms) [3].

Hyperparameter configurations follow the same patterns as the ones we used for the rest of simplicial models. For the Cell MP, we use a final sum readout, 10 layers, and a hidden size of 64.  For the Cellular Transformer, we use the Hodge Laplacian Eigenvectors defined at [2], a tensor diagram where dimension i attends to dimensions i-1 and i+1, and, global mean pooling, a hidden size of 64, 8 heads, and two layers. For the DECT model, we follow [3] and use 32 directions with a resolution of 32 and hence summarize each simplical complex in a 32x32 image. For the classification and prediction we use an MLP with 64 hidden neurons and 3 hidden layers.

Betti numbers:

| Dataset | Transform | Betti 0 Accuracy | | | Betti 1 AUROC | | | Betti 2 AUROC | | |
|----------------|-------------------------|------------------------------|------------------------------|------------------------------|---------------------------|---------------------------|---------------------------|---------------------------------|---------------------------------|---------------------------------|
| | | Cell MP | Cell Transformer | DECT | Cell MP | Cell Transformer | DECT | Cell MP | Cell Transformer | DECT |
| All | Degree Transform | 0.46 ± 0.50 | 1.00 ± 0.00 | 1.00 ± 0.00 | 0.62 ± 0.07 | 0.93 ± 0.01 | 0.50 ± 0.00 | 0.49 ± 0.06 | 0.55 ± 0.00 | 0.50 ± 0.00 |
| All | Degree Transform OneHot | | | 1.00 ± 0.00 | | | 0.50 ± 0.00 | | | 0.50 ± 0.00 |
| All | Random Node Features | 1.00 ± 0.00 | 1.00 ± 0.00 | 1.00 ± 0.00 | 0.84 ± 0.00 | 0.66 ± 0.02 | 0.50 ± 0.00 | 0.52 ± 0.02 | 0.53 ± 0.01 | 0.50 ± 0.00 |
| Known type | Degree Transform | 0.05 ± 0.09 | 1.00 0.00 | 1.00 ± 0.00 | 0.23 ± 0.01 | 0.27 ± 0.01 | 0.21 ± 0.00 | 0.52 ± 0.04 | 0.52 ± 0.03 | 0.50 ± 0.00 |
| Known type | Degree Transform OneHot | | | 1.00 ± 0.00 | | | 0.21 ± 0.00 | | | 0.50 ± 0.00 |
| Known type | Random Node Features | 0.98 ± 0.00 | 1.00 0.00 | 1.00 ± 0.00 | 0.29 ± 0.01 | 0.21 ± 0.00 | 0.21 ± 0.00 | 0.51 ± 0.02 | 0.50 ± 0.00 | 0.50 ± 0.00 |

Homeomorphism type:

| Dataset        | Transform               | AUROC             |                         |                   |
|----------------|-------------------------|-------------------|-------------------------|-------------------|
|                |                         | Cell MP           | Cell Transformer        | DECT              |
| All        | Degree Transform        | 0.85       ± 0.11 | 0.91             ± 0.01 | 0.45       ± 0.00 |
| All        | Degree Transform OneHot |                   |                         | 0.45       ± 0.00 |
| All        | Random Node Features    | 0.89       ± 0.01 | 0.69             ± 0.15 | 0.45       ± 0.00 |
| Known type | Degree Transform        | 0.63       ± 0.14 | 0.83             ± 0.01 | 0.50       ± 0.00 |
| Known type | Degree Transform OneHot |                   |                         | 0.50       ± 0.00 |
| Known type | Random Node Features    | 0.82       ± 0.00 | 0.50             ± 0.02 | 0.48       ± 0.01 |

Orientability:

| Dataset        | Transform               | AUROC        |                  |             |
|----------------|-------------------------|--------------|------------------|-------------|
|                |                         | Cell MP      | Cell Transformer | DECT        |
| All        | Degree Transform        | 0.65  ± 0.07 | 0.55 ± 0.00      | 0.50 ± 0.00 |
| All        | Degree Transform OneHot |              |                  | 0.50 ± 0.00 |
| All        | Random Node Features    | 0.55 ± 0.00  | 0.50 ± 0.00      | 0.50 ± 0.00 |
| Known type | Degree Transform        | 0.51  ± 0.01 | 0.52 ± 0.03      | 0.50 ± 0.00 |
| Known type | Degree Transform OneHot |              |                  | 0.50 ± 0.00 |
| Known type | Random Node Features    | 0.50  ± 0.01 | 0.50 ± 0.00      | 0.50 ± 0.00 |


[1] Bodnar et al. "Weisfeiler and Lehman Go Cellular: CW Networks", NeurIPS 2021.

[2] Ballester, Rubén, et al. "Attending to Topological Spaces: The Cellular Transformer." arXiv preprint arXiv:2405.14094 (2024).

[3 Röell, Ernst, and Bastian Rieck. "Differentiable Euler Characteristic Transforms for Shape Classification." The Twelfth International Conference on Learning Representations.

---

### Author Response · Authors · 2024-11-21
**Summary of the results**

**Summary of experiments:**

*Betti numbers:*

For the Betti numbers, the scores of the DECT model are comparable to the graph-based models.
Regarding the cellular MP and topological  transformer architectures, both obtain close to on-par results to the best models on the respective tasks. For the first Betti number, the cellular MP and transformer architectures obtain better performances than all the networks except for the best simplicial one, SCCN. For the second betti number, they pose scores slightly better than the worst simplicial model but lack behind the best simplicial model by about 8% in average.

*Homeomorphism type:*

For the full set of surfaces, the cell transformer and the cellular message passing network outperform (0.91 ± 0.01 and 0.89 ± 0.01, respectively) the previous best AUROC of SCCN (0.85 ± 0.08). The DECT model obtain worse values than simplicial models, in general, having results more similar to the ones of the graph networks.

For the set of surfaces with homeomorphism type assigned the same happens: the AUROC metrics are 0.83 ± 0.01 and 0.82 ± 0.00, respectively for cellular transformer and message passing architectures, and the best previous score was (0.80± 00) for SCCN.

*Orientability:*

For the full set of surfaces, the cellular MP outperforms the previous best models (0.65± 0.07 against 0.55 ± 0.01 and 0.55± 0.09 for SCCN and SCCNN, respectively). The cellular transformer matches the same performance (0.55 ± 0.00).

For the set of surfaces with homeomorphism type assigned, both, cellular transformers and cellular message passing networks obtain comparable performances to most most of the other networks, except for SCCN, where they are slightly worse (0.52± 0.03 and 0.51 ± 0.01 for transformer and cell MP against 0.54 ± 0.01 for SCCN, respectively).

**Conclusion:**

Although we saw that for the Betti numbers' experiments the new models did not perform better than the older ones, we also saw that the cellular MP and topological transformer architectures obtained slightly better results than the older models for most homeomorphism type and orientability experiments, being comparable in the ones they were not better. Still, the differences are minimal and thus we believe they do not change our conclusions for the analysis on the MANTRA surfaces.

---

### Meta-Review · Area_Chair_ZRsH · 2024-12-22

**Metareview:**

The paper proposes MANTRA, mainly a synthetic new benchmark focusing on evaluating methods in topological deep learning. Most reviewers were on the positive side, although there is quite a big disagreement with a 'accept' and a 'reject' suggestion. Basically, the point of disagreement is to what extent the paper presents a truly new data setting and benchmark, given that similar and harder problems have been extensively studied and researched in graphics, specifically CGAL (computational geometry algorithms library). The counterargument is that these libraries are indeed addressing a similar problem, although it is definitely non-trivial for researchers who come from ML and are not domain experts to work with these libraries. This is certainly a logical point.

Another point of concern is whether having a dataset like MANTRA is sufficient, given that it is a synthetic dataset, and given that there is a plethora of real-world problems that could be considered as well.

I believe both sides have merit and I am definitely sensitive to the argument that it is important that ML methods stay close to real-world and practical applications. To decide on this, I note that since this is a dataset+benchmark paper, ignoring the 'magnitude' of the recommendations and only focusing on the 'sign' (positive vs negative review), most reviewers agree it is a useful dataset. Combined with the fact that as it was commented by the reviewers, the papers addresses a big gap in TDL, I recommend acceptance.

**Additional Comments On Reviewer Discussion:**

Reviewers in the post-rebuttal phase noted differing perspectives on the contribution of the dataset proposed in the paper. On one hand, it was highlighted that existing higher-order topology datasets, such as those in CGAL and real-world networks, already address aspects of triangulations and higher-order relations, questioning the novelty and applicability of the proposed dataset, which focuses only on synthetic 2D and 3D shapes. Concerns were also raised about insufficient benchmarking and experimental setups, with suggestions for broader baselines and more robust input labels. On the other hand, it was argued that the dataset uniquely enumerates purely topological simplicial complexes in a controlled manner, distinguishing it from CGAL-generated datasets, which rely on geometric relations. While this unique aspect was acknowledged as valuable for early-stage research in TDL, reviewers expressed concerns about its limited real-world applicability and long-term relevance, emphasizing the need to bridge the gap between synthetic and practical datasets.

---

### Decision · Program_Chairs · 2025-01-22

Accept (Poster)